# The inhibitory microcircuit of the substantia nigra provides feedback gain control of the basal ganglia output

Jennifer Brown[1,2], Wei-Xing Pan[1], Joshua Tate Dudman[1]*

[1]Janelia Farm Research Campus, Howard Hughes Medical Institute, Ashburn, United States; [2]Department of Physiology, Development and Neuroscience, University of Cambridge, Cambridge, United Kingdom

**Abstract** Dysfunction of the basal ganglia produces severe deficits in the timing, initiation, and vigor of movement. These diverse impairments suggest a control system gone awry. In engineered systems, feedback is critical for control. By contrast, models of the basal ganglia highlight feedforward circuitry and ignore intrinsic feedback circuits. In this study, we show that feedback via axon collaterals of substantia nigra projection neurons control the gain of the basal ganglia output. Through a combination of physiology, optogenetics, anatomy, and circuit mapping, we elaborate a general circuit mechanism for gain control in a microcircuit lacking interneurons. Our data suggest that diverse tonic firing rates, weak unitary connections and a spatially diffuse collateral circuit with distinct topography and kinetics from feedforward input is sufficient to implement divisive feedback inhibition. The importance of feedback for engineered systems implies that the intranigral microcircuit, despite its absence from canonical models, could be essential to basal ganglia function.

*For correspondence: dudmanj@janelia.hhmi.org

**Competing interests:** The authors declare that no competing interests exist.

**Reviewing editor**: Sacha B Nelson, Brandeis University, United States

## Introduction

The basal ganglia are a collection of interconnected subcortical regions of the vertebrate brain (**DeLong, 2000**). Pathological disruptions of basal ganglia signaling produce profound deficits in the timing (**Buhusi and Meck, 2005**), vigor (**Turner and Desmurget, 2010**), and initiation (**Mink, 1996**) of voluntary movements. While it is thus clear that the basal ganglia are critical for voluntary movement, the specific mechanisms by which movement is controlled by basal ganglia activity remain unclear (**Turner and Desmurget, 2010**). Voluntary control of movement can be explained in terms of optimal feedback control theory (**Diedrichsen et al., 2010**). The basal ganglia circuit can be described as an extended loop that begins with projections from deep layer cortical neurons and ultimately returns to the cortex via projections from the basal ganglia to the ventral thalamus (**Haber, 2003**). However, the basal ganglia circuit also contains intrinsic feedback projections (**Gerfen, 2004**). In engineered control systems, feedback is critical for stable performance (**Astrom and Murray, 2008**).

The substantia nigra (SN) pars reticulata (SNr) is the primary source of output from the sensorimotor basal ganglia in rodents (**Gerfen, 2004**). The vast majority of neurons in the SNr are projection neurons that synthesize and release the neurotransmitter Υ-aminobutyric acid (GABA). Projection neurons of the SNr target pre-motor areas in the ventral thalamus, dorsal midbrain, and tegmentum (**Parent, 1990**; **Hikosaka, 2007**). In addition to these long range targets, nigral projection neurons also elaborate axon collaterals within the SN (**Mailly et al., 2003**; **Cebrián et al., 2005**; **Deniau et al., 2007a**). There are no known interneurons in the SNr (**Deniau et al., 2007a**), and thus collaterals of projection neurons are the sole source of intrinsic feedback for the basal ganglia output. Anatomical reconstructions have indicated that the axon collaterals of SNr projection neurons are sparse (**Mailly et al., 2003**). The functional impact of this intranigral microcircuit remains unclear. Antidromic activation of

**eLife digest** The basal ganglia are a group of nuclei located deep within the brain that are involved in the control of movement. The death of neurons in one particular nucleus—known as the substantia nigra—gives rise to a range of symptoms that are characteristic of Parkinson's disease, including slowness of movement and tremors.

Although the basic anatomy and circuitry of the basal ganglia were worked out many years ago, it is not clear how these structures control voluntary movement. Based on insights from engineering, Brown et al. propose a model in which negative feedback within the substantia nigra—largely overlooked by previous models—regulates the output of the basal ganglia and thus contributes to the control of movement.

Most areas of the brain contain projection neurons, which connect to other areas of the brain, and interneurons, which do not form connections beyond the nucleus in which they reside. In these areas, dedicated networks of interneurons use feedback to exert control over the signals that the projection neurons carry to other areas of the brain.

However, it is thought that the substantia nigra does not contain interneurons. This led Brown et al. to propose that structures called axon collaterals form a microcircuit that can instead supply such feedback in the substantia nigra. Axons are the nerve fibres that carry signals away from the cell body of a neuron, and axon collaterals are branches of those axons. Data obtained by recording and manipulating electrical activity in the substantia nigra were consistent with this model and further experiments allowed this microcircuit to be mapped in detail.

By revealing the circuit mechanisms of negative feedback within the substantia nigra, the work of Brown et al. changes our understanding of the basal ganglia and could have implications for understanding the mechanisms and ultimately the treatment of disorders such as Parkinson's disease.

SNr projection neurons in anesthetized animals has been used to infer the presence of inhibition via projection neurons collaterals (*Tepper and Lee, 2007*; *Brazhnik et al., 2008*); however, the relative impact, spatial organization and temporal properties of recruitment of feedback inhibition via SNr collateral inhibition remains largely unknown. In this study, we explore the hypothesis that the microcircuit formed by SNr collaterals could implement a critical negative feedback node in the context of a control system for voluntary behavior that is implemented in the extended cortico-basal ganglia circuit.

In engineered systems, the functional impact of a negative feedback can be difficult to detect and characterize (*Astrom and Murray, 2008*). For example, at steady state or in the absence of change in the state of the system, there may be no obvious effect of appropriately functioning negative feedback. However, sudden transitions in the state of the system can reveal the contribution of negative feedback—altering, for example, the gain and/or the time course of settling around transitions. By analogy to an engineered system, the SNr microcircuit may have little apparent impact in the absence of sudden changes in the state of the population activity. However, changes in behavior and receipt of sensory stimuli are accompanied by phasic transitions, both increases and decreases, in the activity of the SNr population. We thus reasoned that the role of negative feedback, implemented by the SNr microcircuit, could become apparent under such conditions. Recent work has shown that a salient or conditioned stimulus (CS), for example an auditory tone, can lead to phasic changes in the activity of SNr neurons in rodents (*Schmidt et al., 2013*). Moreover, these short-latency modulations of activity are predictive of the initiation of conditioned behavioral responses—that is action selection (*Pan et al., 2013*; *Schmidt et al., 2013*). If the basal ganglia act as a control system for behavior, then we would predict that control over the gain or dynamic range of these phasic modulations should be critical for normal voluntary actions.

Detailed study of the local inhibitory connections within the SNr has been hampered by the difficulty in isolating and specifically exciting the SNr collaterals independent of afferent inhibitory and excitatory projections (*Hammond et al., 1983*). We overcame this challenge by using cell-type specific expression of channelrhodopsin-2 (ChR2), a light-gated cation channel (*Boyden et al., 2005*; *Zhang et al., 2006*), in SNr GABA neurons. This optogenetic approach allowed us to stimulate SNr GABA neurons with high temporal and spatial resolution without contamination from excitatory afferents, dopaminergic transmission, or afferent inhibitory input from the striatonigral projection both in vitro and in vivo.

Consistent with the prediction from anatomical data, we show that inhibition derived from the collaterals of projection neurons in the SNr is sparse and has little to no effect on tonic baseline firing. However, we also observed that activation of the SNr projection neuron population could elicit a large and potent feedback inhibition capable of shaping output activity. Here, we show that this unique combination of effects is the result of a number of distinctive features of the SN microcircuit: (1) post-synaptic currents resulting from collateral synaptic input provided robust inhibition with a rapid onset during strong activation of the network; (2) unitary connections are weak with sufficiently low release probability to sustain release during repetitive stimulation; (3) asynchronous basal inhibition in the tonically active network is effectively compensated for by intrinsic conductances that sustain tonic spiking; (4) the potency of feedback inhibition is proportional to total activation of the microcircuit due to a sparse, spatially diffuse connectivity. Together, these properties of the intrinsic microcircuit of the SNr implement a robust inhibition that is rapidly and stably recruited in proportion to the sustained activation of the projection neuron population with little effect in the absence of stimulation—in other words, the inhibitory microcircuit of the SNr mediates a divisive gain control on the basal ganglia output.

## Results

If collateral, feedback projections of SNr neurons provides a source of negative feedback, then we would expect that the level of activation of the population immediately prior to a stimulus should be inversely correlated with the modulation of the response of an individual neuron to that stimulus. In other words, if there were more activity producing collateral inhibition, the phasic response to a stimulus could be blunted. To test this possibility, we examined a dataset of 599 single units recorded from the ventral midbrain of mice ($n = 5$, strongly biased towards recordings from GABAergic neurons in the SN, 'Materials and methods') performing a classical trace conditioning task described previously (**Pan et al., 2013**). For each individual unit, we computed the normalized response to a salient stimulus (an auditory tone that predicted a delayed reward) as a function of the normalized activity of the simultaneously recorded population of neurons for 32 recording sessions in which at least eight neurons were recorded simultaneously (median = 12; maximum = 21) for each trial in the recording session (median = 71 trials; range = 42:132) (**Figure 1A–B**). This yielded a data set of 28,277 comparisons from which we estimated the correlation between the activation of the population at baseline to the activation of each individual neuron in response to the conditioned stimulus (CS). We found that both the

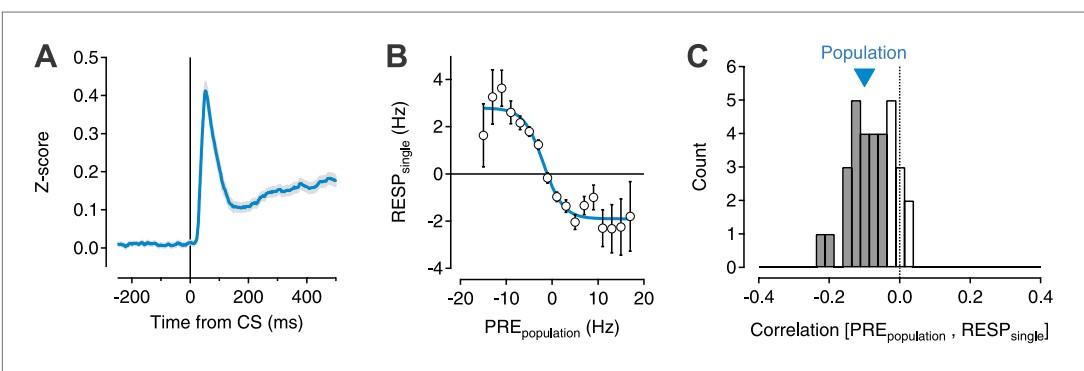

**Figure 1**. Transient changes in the basal ganglia output are reduced by ongoing population activity. A data set of 599 single units was isolated from recording sessions with at least eight simultaneously recorded units (**Pan et al., 2013**). Electrodes were targeted to the substantia nigra and ventral tegmental region of mice trained to perform an auditory trace conditioning paradigm. (**A**) Spiking activity was z-scored, aligned to the onset of the conditioned stimulus (CS), and averaged for all units. (**B**) For each recorded unit the mean subtracted response (RESP$_{single}$) was computed as a function of the mean normalized activity prior to CS onset for the rest of the simultaneously recorded population (PRE$_{population}$; 7–20 units). Population data were binned, averaged, and fit with a sigmoid function (cyan line). (**C**) The correlation coefficient between RESP$_{single}$ and PRE$_{population}$ was computed for each session ($n = 32$). A histogram of all correlation scores is drawn with significant correlations (permutation test) indicated by filled gray bars. The correlation score of the entire population (−0.1) is indicated by a cyan triangle.

population (−0.1; p<0.001 permutation test) and 22/32 individual sessions exhibited significant negative correlations (−0.12 ± 0.05 SD; p<0.05, permutation test; *Figure 1C*).

These data thus suggest that the activation of the SNr population indeed provides a negative feedback to limit the phasic response of the population to salient stimuli. However, these data also imply a surprising structure to the SNr microcircuit—namely, even a relatively poor estimate of population activity (7–20 simultaneously recorded units across electrodes spread over hundreds of microns) is sufficient to provide predictive power for the response of individual units. However, significant individual pairwise correlations were very rarely observed in these recordings (*Pan et al., 2013*) consistent with prior work (*Nevet et al., 2007*). This would suggest that either the negative correlation observed is a consequence of correlations in activity due to extended feedback projections or that individual units receive relatively weak and diffuse input from many SN neurons rather than strong feedback inhibition from proximally located neurons.

To distinguish between these possibilities, we first sought to test whether this negative feedback property of the SN microcircuit could be recapitulated in an in vitro preparation where extrinsic, multisynaptic sources of feedback are eliminated. The functional properties of feedback inhibition to the basal ganglia were assessed by channelrhodopsin-2 (ChR2) mediated stimulation of SNr GABA neurons. In one set of experiments an adeno-associated virus (AAV) that expressed a *cre*-dependent ChR2 transgene (*Atasoy et al., 2008*) was injected into the SNr of a mouse line in which *cre*-recombinase was expressed under the glutamic acid decarboxylase (*Gad2*) promoter to target expression to SNr GABA neurons (hereafter referred to as *Gad2*-ChR2; *Figure 2A,C*). In the other, we exploited a transgenic mouse line which has strong expression of ChR2 under the control of the thymus cell antigen 1 (*Thy1*) promoter (*Arenkiel et al., 2007*), (hereafter referred to as *Thy1*-ChR2; *Figure 2B,D*). In this transgenic line ChR2 is robustly expressed in SNr GABA neurons, but not in SN dopaminergic neurons (*Pan et al., 2013*;) or in upstream projection neurons of the dorsal striatum (data available from the authors on request). Both approaches thus provide a method to specifically excite SNr GABA neurons with high reliability and fine temporal resolution (*Figure 2E–J*).

## Local SNr inhibition is sufficient to modify basal ganglia output during phasic activation

Local axon collaterals of projection neurons provide a source of feedback inhibition proportional to the output of the SNr. For this inhibition to regulate the output of the SNr it must be sufficient to suppress activity even in the presence of strong, phasic activation of the projection neurons. Phasic activation of the SNr population occurs, for example, at the onset of salient sensory cues (*Pan et al., 2013*; *Schmidt et al., 2013*; *Figure 3—figure supplement 1*). Thus, to determine whether local inhibition was sufficient to regulate the gain of the SNr network, we used ChR2 stimulation to drive repetitive somatic spiking in the projection neuron network. This recruits a population of SNr neurons with a time course and distribution of responses similar to that evoked by conditioned stimuli (*Pan et al., 2013*; *Figure 3—figure supplement 1*). We determined the consequences of local inhibition by comparing activity evoked when inhibition was intact with activity evoked following pharmacological blockade of inhibition.

Whole cell current-clamp recordings from individual SNr projection neurons were obtained from brain slices of *Thy1*-ChR2 mice in the presence of excitatory synaptic transmission blockers (D-AP5 and NBQX; *Figure 3A*). Wide-field illumination through a 10X objective was used to stimulate activity throughout the SNr network. Direct light-evoked spiking in the recorded neuron was substantially, or in some cases completely, suppressed under control conditions (*Figure 3B,C*). However, reliable light-evoked spiking was always present following application of the GABA$_A$ receptor antagonist gabazine (Gbz) to block local inhibition (*Figure 3D*). The suppression of evoked spiking was consistent across stimulus durations within a cell (*Figure 3E*), whereas the magnitude of suppression was more idiosyncratic across cells for a given stimulus condition (*Figure 3F*).

The ability of feedback inhibition to suppress spiking more effectively with increasing stimulus duration (*Figure 3E*) implies a divisive gain control. To quantify the gain effect across the population, we compared the normalized response to photostimulation of increasing duration both in the presence and absence of inhibitory synaptic transmission (*Figure 3G*). We found that the response of the population showed a significant increase as a function of stimulus duration and the magnitude of the increase was significantly reduced by the presence of feedback inhibition (two-factor ANOVA; p<0.05). Divisive gain control is characterized by a suppression of spiking at large stimuli but little to

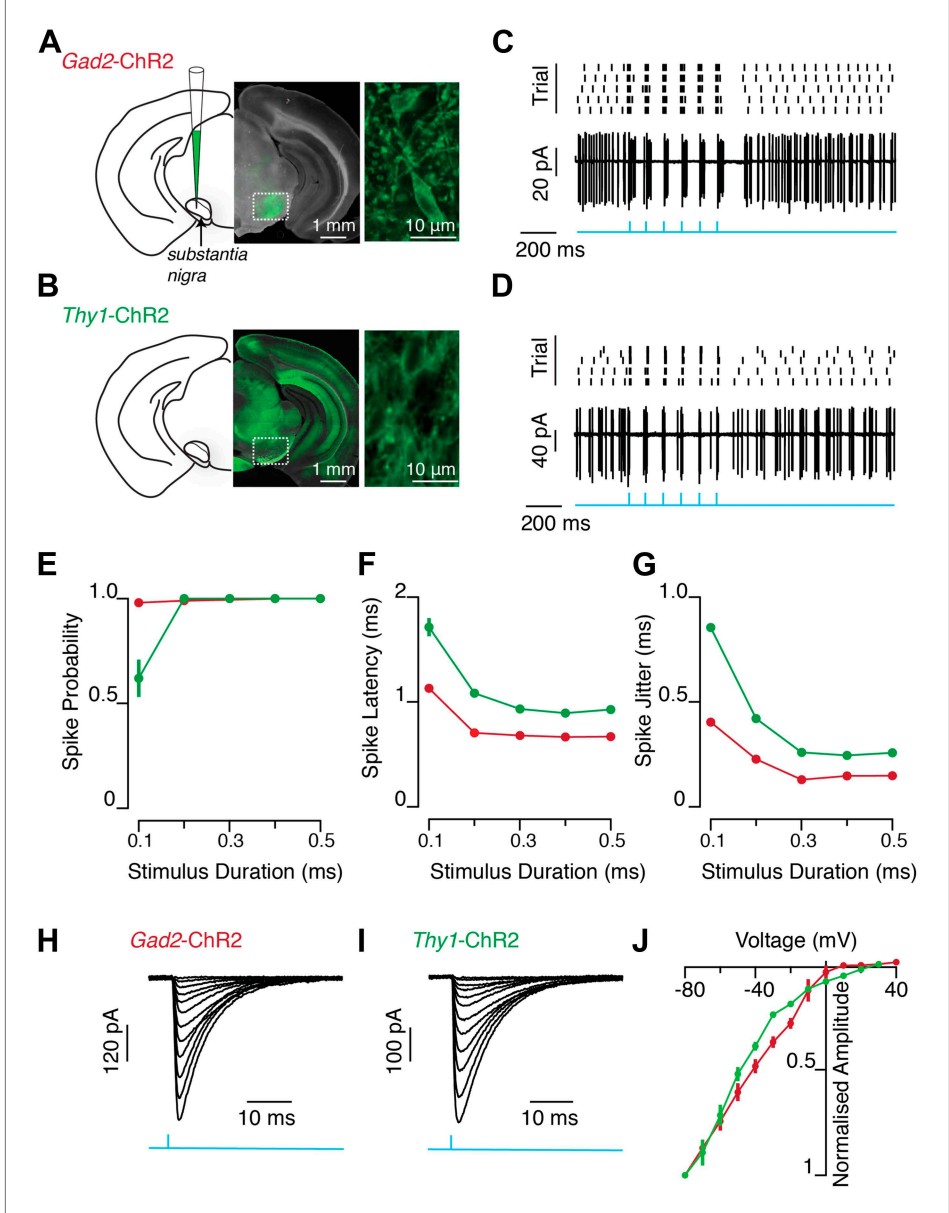

**Figure 2**. Light evoked activity of ChR2-expressing SNr GABA neurons in vitro. ChR2 was selectively expressed in SNr GABA neurons via two methods. Viral injection of AAV expressing *cre*-dependent ChR2-GFP transgene into SNr of a mouse line in which *cre*-recombinase was expressed under the glutamic acid decarboxylase (*Gad2*) promoter (**A**, *Gad2*-ChR2) and transgenic mouse line (*Thy1* Line18) which has ChR2 expression under the control of *Thy1* promoter (**B**, *Thy1*-ChR2), (**A**–**B**) left: schematic of midbrain region with SN labeled and pipette representing injection target in (**A**), middle: midbrain coronal sections showing ChR2-GFP expression (green), right: two-photon image of ChR2-GFP positive SNr GABA neurons. In vitro wide-field illumination of midbrain slice (0.5 ms light pulse, 10 Hz, cyan arrows) reliably evoked action potentials in SNr GABA neurons in *Gad2*-ChR2 (**C**; n = 12/20 cells) and *Thy1*-ChR2 (**D**; n = 21/21 cells) mice, (**C** and **D** rater plot [upper] and cell-attached recording [lower] of a representative neuron from each mouse line showing evoked spiking over five trials repeating the same light stimulus. Quantification of light evoked spiking probability (**E**), latency (**F**) and standard deviation of the latency (jitter) (**G**) for a range of photostimulation durations recorded from both *Gad2*-ChR2 (red; n = 5 cells) and *Thy1*-ChR2 (green; n = 7 cells) mice. Representative light evoked ChR2-mediated inward current recorded at a range of membrane voltages (from −80 mV to +40 mV) from a single neuron recorded in either the *Gad2*-ChR2 (**H**) or *Thy1*-ChR2 (**I**) mouse. (**J**) Current-voltage relationship of light-evoked currents recorded in either *Gad2*-ChR2 (red; n = 8 cells) or *Thy1*-ChR2 (green; n = 8 cells) mice.

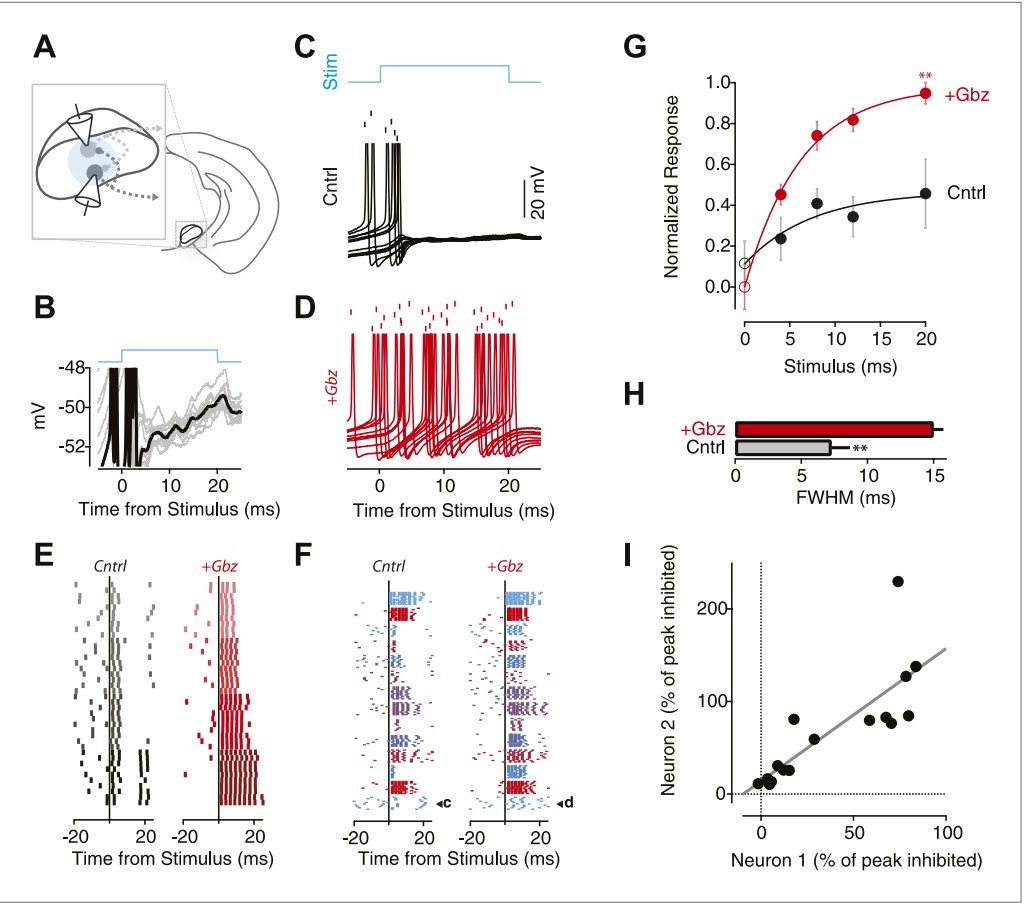

**Figure 3**. The local inhibitory microcircuit of the SNr provides feedback gain control. (**A**) Schematic of the experimental configuration. 1-2 SNr GABA neurons were recorded from in the whole-cell current clamp configuration. Wide-field illumination of the slice (indicated by cyan circle) was used to photostimulate the SNr network. (**B**) Example recording from an individual SNr GABA neuron during light stimulation (upper cyan trace). Note the stereotyped membrane potential fluctuations during photostimulation (10 trials). (**C** and **D**) Example recordings from the same neuron recorded during 10 trials of stimulation ('Stim'; upper cyan trace) under control conditions (**C**; Cntrl; black) and following pharmacological blockage of inhibition via gabazine application (**D**; +Gbz; red) aligned to stimulus onset. Tick marks indicate spike times for 10 repetitions of the same light stimulus. Lower traces show the intracellular recording from the same neuron overlaid for all trials. (**E**) Raster plot of evoked spiking in control conditions (left) and in the presence of Gbz (right) for 4 blocks of 10 trails of increasing stimulus durations (4 ms, 8 ms, 12 ms, 20 ms; top to bottom) for a single neuron. (**F**) Raster plots of evoked spiking for 10 trials aligned to the onset of an 8 ms light stimulus for the population of neurons under control conditions (left) and in the presence of Gbz (right). (**G**) Normalized response across the population of neurons binned by stimulus duration and grouped by treatment (black, Cntrl and red, +Gbz). Significant effects on both stimulus duration and treatment condition were observed (Two-way ANOVA, $p < 0.05$). Zero stimulus responses (open symbols) were estimated from the background firing rate. No significant difference was observed. (**H**) Full width half maximum (FWHM) of the peristimulus time histogram (PSTH) for evoked spiking in control and following Gbz ($p < 0.01$). (**I**) For paired recordings, the percent inhibition of one neuron in the pair was plotted as a function of the percent inhibition of the other neuron for all stimulus conditions (black circles). A significant positive correlation was found and indicated by the solid black line ($p < 0.01$; two tailed $t$ test).

The following figure supplements are available for figure 3:

**Figure supplement 1**. Direct comparison of responses elicited by optogenetic and natural stimulation.

**Figure supplement 2**. mIPSC amplitude in SNr GABA neurons.

no effect on the response to weak or absent stimuli. Consistent with a divisive gain control, we found that baseline firing was unaffected by removal of inhibition (*Figure 3G*, open circles). To contrast a feedback gain control with the effect predicted for subtractive inhibition, we examined recordings from dopamine neurons of the SN that do not express ChR2, but are strongly inhibited by ChR2 expressing projection neurons (*Pan et al., 2013*). For dopamine neurons, we observed a constant suppression of spiking across the range of stimuli used (*Pan et al., 2013*). If the reduction in spiking observed during stimulation was the result of feedback inhibition one would predict that the inhibition should onset after the onset of the population response and truncate the response present in the absence of feedback. Consistent with this prediction we found that suppression of spiking was characterized by a significant decrease in the duration of the evoked spiking (*Figure 3H*). This effect on the duration could be observed in many individual neuron responses (*Figure 3F*).

Our in vivo results suggested that the extent of suppression of transient activation in SNr neurons is proportional to the estimated activation of the network (*Figure 1*). Anatomical studies indicate that the vast majority of projection neurons elaborate collaterals within the SNr, however, these collaterals can form relatively few (~10) putative synaptic contacts (*Mailly et al., 2003*). Moreover, we found that unitary release events produced relatively weak mIPSCs (~150 pS) (*Figure 3—figure supplement 2*). Taken together, these data imply that the inhibition observed results from the activation of approximately 50–100 presynaptic inputs. Given that the GABAergic neurons in the SNr are thought to be exclusively projection neurons (*Deniau et al., 2007b*), this is consistent with the finding that projection neuron collaterals form 79.4 ± 96.1 (SD) boutons per neuron within the SNr of the rat (*Mailly et al., 2003*).

Assuming a modest or low probability of paired connections, our connectivity estimates imply that the extent of activation across a large population of SNr neurons would determine the extent of feedback inhibition consistent with our observation in vivo. While we do not have a direct measure of the total extent of activation of the SNr by our photostimulation, we note that neurons recorded simultaneously experience the same activation state of the network. Thus, we reasoned that the extent of feedback inhibition in a pair of recorded neurons should be correlated if feedback inhibition is proportional to the total activation of the network. Consistent with this prediction, we found that there was a significant correlation (Pearson's correlation, $p<0.01$ permutation test) in paired recordings (*Figure 3I*). These results suggest that a given SNr projection neuron receives input from a spatially diffuse collection of other SNr projection neurons.

The data above are consistent with the claim that a large population of SNr projection neurons must be recruited to fire within a relatively small time window (5–20 ms) in order to achieve robust feedback inhibition and divisive gain effects (*Figure 3G*). These results were obtained in the *Thy1*-ChR2 mouse where all neurons in the SNr express ChR2 (*Figure 2*). This would suggest that if a local subset of the SNr was expressing ChR2, the divisive gain effect should be present, but reduced in magnitude analogous to the smaller effects observed when less of the network was recruited in the *Thy1*-ChR2 preparation (*Figure 3G*). Indeed, we found that when the same experiment was repeated in ChR2+ nigral neurons from virally infected *Gad2*-ChR2 mice a divisive gain effect was observed, but reduced in magnitude ($p<0.05$; two-factor ANOVA; 15% reduction in the saturated response).

## Intrinsic properties that produce tonic spiking effectively counteract transient inhibition

To alter the gain of a response to activation of the network requires a change in the slope of the curve. As described above, we observed that there was a substantial effect of feedback inhibition in the strongly activated SNr circuit, but no effect in the absence of stimulation—resulting in a change in the slope of the response to stimulation. However, it is confusing how a strongly coupled inhibitory network of tonically active neurons could exhibit no effect of feedback even in the absence of stimulation. We first asked whether the rate of spontaneous IPSCs was consistent with our estimate, and a prior anatomical estimate (*Mailly et al., 2003*), of >50 inputs from other SNr projection neurons. The expected rate of spontaneous IPSCs would thus be approximately:

$$R_{uIPSCs} = N_{pre} \times R_{pre} \times P_{release} \tag{1}$$

where, $R_{uIPSCs}$ is the predicted rate of unitary IPSCs (uIPSCs), $N_{pre}$ is the number of presynaptic inputs (release sites), $R_{pre}$ is the mean firing rate of presynaptic inputs, and $P_{release}$ is the effective release probability across all release sites. Thus, with a relatively low release probability (<0.5), we would predict

75–300 Hz of uIPSCs. This corresponds well to the rate of uIPSCs estimated directly from voltage clamp recordings (*Figure 4A–C*). Consistent with this estimate, we also show that repetitive stimulation of SNr collaterals fails to completely depress transmission (*Figure 4—figure supplement 2*) consistent with a vesicular release probability low enough to allow vesicle recycling to keep pace with release. Such a mechanism has been described in detail for Purkinje cell synapses (*Telgkamp et al., 2004*). These observations suggest that there is indeed a substantial background rate of IPSCs that, when pharmacologically blocked, has no significant effect on the tonic firing of SNr projection neurons (*Atherton and Bevan, 2005*).

The question as we posed it—how can a strongly coupled inhibitory network of tonically active neurons exhibit no effect of feedback under basal conditions?—implies that tonic spiking is the problem. Alternatively, tonic spiking could be the solution. For a neuron to repetitively fire it must, upon the return from a spike, exhibit a net membrane current that is inward and thus drives the membrane towards spike threshold (*Raman and Bean, 1999*). This implies a positive slope conductance

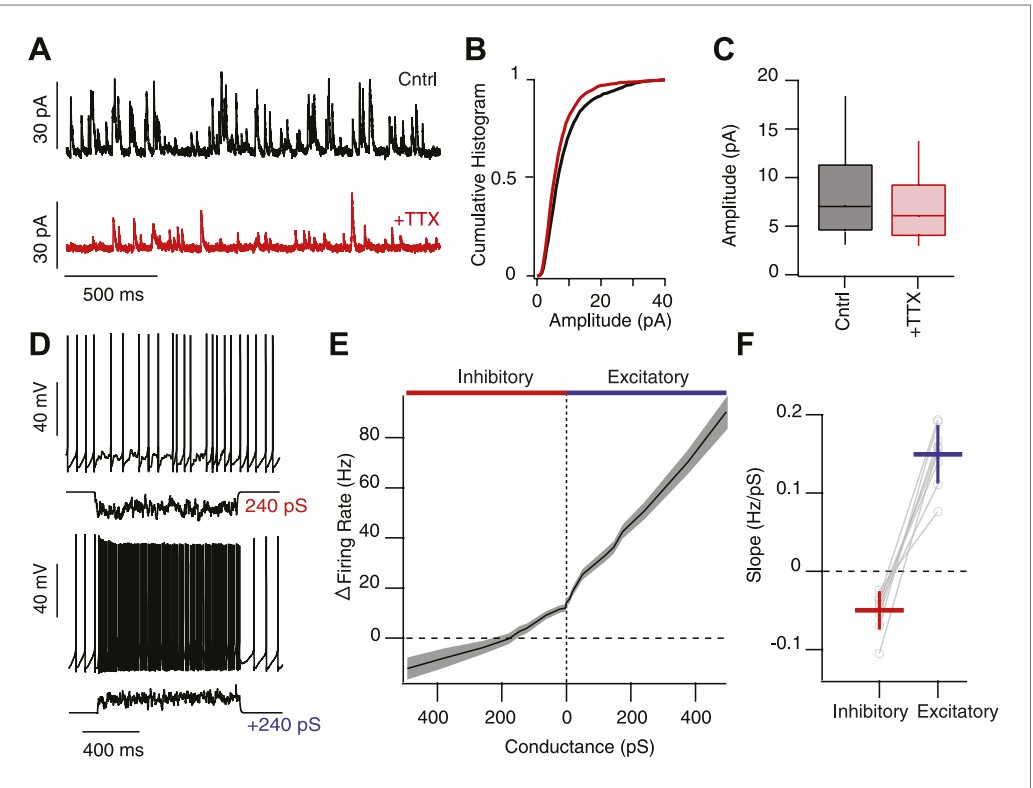

**Figure 4**. High background inhibition has little affect on tonic activity of SNr neurons. (**A**) Whole-cell recording of spontaneous IPSCs (sIPSCs) onto SNr neurons in control conditions (Cntrl; black trace) and following addition of tetrodotoxin (TTX) to isolate miniature events (+TTX; red trace, $V_h$ 0 mV). (**B**) Cumulative histogram of IPSC amplitude in control conditions and following addition of TTX ($n = 4$ cells). (**C**) Box and whisker plot of IPSC amplitude for control and following addition of TTX. (**D**) Spiking output of SNr neurons following addition of high background excitation (upper) or inhibition (lower) via the dynamic clamp. (**E**) Summary data of change in firing rate of SNr neurons ($n = 11$ cells) following an increasing the relative frequency of inhibitory (red) or excitatory conductances (blue). (**F**) The slope of the change in firing rate as a function of change in conductance was significantly greater following increases in excitatory conductance compared to inhibitory conductance.

The following figure supplements are available for figure 4:

**Figure supplement 1**. Intrinsic, net inward currents and a positive slope conductance allows feedback gain control of SNr neurons.

**Figure supplement 2**. Low release probability and sustained depression at feedback inhibitory synapse.

combined with a net inward current below threshold (*Nolan et al., 2003*)—in other words, the conductances that drive repetitive firing oppose hyperpolarizing currents in the perithreshold regime. Combined with a reduced driving force of inhibition near threshold, these biophysical features suggest that SNr neurons are much less sensitive to inhibition than to excitation in this regime. To test this hypothesis explicitly, we performed dynamic clamp experiments in which we systematically varied the balance between a high background rate of IPSCs and EPSCs (*Figure 4D–F*). Indeed, we found that the sensitivity of the spike rate to increasing inhibition was much reduced compared to the sensitivity to increasing excitation. Stimulation strongly biased towards an inhibitory conductance often exhibited no effect on the mean spike rate relative to tonic levels. Consistent with the mechanistic model described above, we found that the slope conductance in the perithreshold regime was indeed nonlinear with a sharp positive slope near the inhibitory reversal potential (*Figure 4—figure supplement 1*). Moreover, we found that measured biophysical properties (e.g., slope conductance, spike threshold) were unaffected by pharmacological blockade of inhibition (*Figure 4—figure supplement 1*).

## The intrinsic inhibitory microcircuit provides a fast, transient inhibition

Collateral inhibition resulted in a strong suppression of evoked spiking and was sufficient to truncate evoked responses, often after only a few milliseconds. This suggested that collateral inhibition provided substantial inhibition that onset rapidly following stimulation. However, it is possible that the transient effect could also reflect properties of the photostimulation. To distinguish these possibilities, we examined the kinetics of feedback inhibition and compared it to the main source of feed forward inhibition to the SNr from the striatum.

We made intracellular recordings from individual SNr projection neurons in *Gad2*-ChR2 mice to probe the properties of local feedback inhibition (*Figure 5A,B*) and from *Drd1a*-cre mice which were injected with a virus expressing a *cre*-dependent ChR2 transgene into the striatum to target the D1 receptor expressing medium spiny neurons which send axons directly into the SNr (*Gerfen, 1988*) (*Drd1a*-ChR2; *Figure 5C*). Postsynaptic neurons were recorded in the voltage clamp configuration with a holding potential of ~+20 mV (reversal potential of the ChR2 current; *Figure 2H–J*) to isolate inhibitory postsynaptic currents (IPSCs). Slices were perfused with antagonists of excitatory synaptic transmission. Repeated pulses (10 Hz) of wide-field photostimulation elicited stimulus locked IPSCs in 15/23 SNr GABA neurons in the *Gad2*-ChR2 mouse line (*Figure 5B*) and 18/24 SNr GABA neurons in the *Drd1a*-ChR2 mouse (*Figure 5C*). Outward currents recorded following photostimulation of both inputs were completely abolished by application of Gbz (*Figure 5B,C*; p<0.001). Evoked IPSCs from feedback nigral collaterals recorded in *Gad2*-ChR2 mouse exhibited rapid kinetics characterized by short, monosynaptic latencies (*Figure 5D*; 1.93 ± 0.02 ms), rapid 10–90% rise times (*Figure 5E*; 0.53 ± 0.01 ms) and rapid decay time constants (τ) (*Figure 5F*; 5.64 ± 0.14 ms). In contrast to the intranigral inhibitory synapses, we found that striatonigral IPSCs recorded in *Drd1a*-ChR2 mouse had significantly longer latencies (*Figure 5D*; 2.51 ± 0.02 ms p<0.001), slower 10–90% rise times (*Figure 5E*; 0.75 ± 0.014 ms, p<0.001) and slower decay τ (*Figure 5F*; 9.00 ± 0.14 ms, p<0.001). Perhaps most surprisingly, we observed that with saturating stimulation, intranigral synapses contributed as large or greater inhibition than the major afferent source of inhibitory input, the direct pathway (*Figure 5G*).

To probe the short-term plasticity properties of feed forward (*Drd1a*-ChR2) and feedback (*Gad2*-ChR2) inhibition to the SNr, we analyzed the amplitude of successive IPSCs. We found that collateral synapses within the SNr exhibited paired pulse ratios (PPR) less than 1 (*Figure 5H*; PPR = 0.91 ± 0.03). While, in direct contrast, the striatonigral synapse was modestly facilitating (*Figure 5H*; PPR = 1.17 ± 0.05). This latter observation was consistent with a previous study that used extracellular stimulation of the direct pathway (*Connelly et al., 2010*). The depressing nature of the PPR for local feedback inhibition is unlikely to reflect desensitization of ChR2 as repeated pulses were all suprathreshold under our stimulus conditions (*Figure 2*). These results are thus consistent with a rapid onset of feedback inhibition sufficient to truncate sustained activation of the SNr population.

We did not find any statistically significant differences between the kinetic properties seen between the IPSCs measured in slices from *Gad2*-ChR2 mice compared with those measured from *Thy1*-ChR2 mice (*Figure 5—figure supplement 1*). This is consistent with both approaches selectively or predominantly activating SNr projection neurons. By contrast, the amplitude of IPSCs evoked by maximal stimulation was significantly reduced in slices taken from *Gad2*-ChR2 mice compared to those taken from *Thy1*-ChR2 mice (p<0.05; unpaired two-tailed *t* test; *Figure 5—figure supplement 1*). These

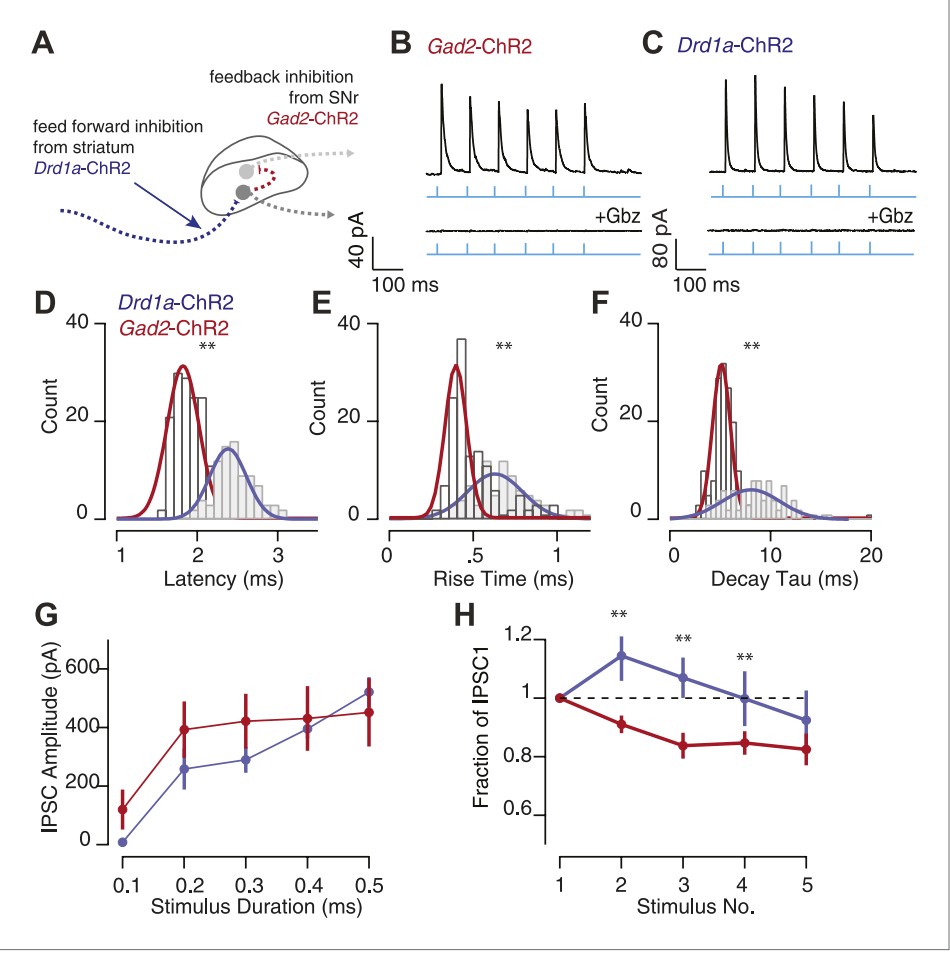

**Figure 5**. Feedback inhibition has distinct biophysical properties from feed forward inhibition. (**A**) Schematic showing feedback nigral synapse (red arrow; *Gad2*-ChR2) and feed forward striatonigral synapse (blue arrow; *Drd1a*-ChR2) onto SNr GABA neurons. Synaptic properties of feedback inhibition were compared to feed forward inhibition to the SNr using photostimulation of ChR2 expressing SNr and striatal axons respectively. Wide-field 10 Hz photostimulation (cyan) to evoked activity of SNr GABA neurons in the *Gad2*-ChR2 mouse elicited large feedback IPSCs in SNr GABA neurons that were blocked with Gbz (**B**; *Gad2*-ChR2, n = 6 cells, p<0.001, paired two tailed *t* test). Similarly, photostimulation of striatonigral afferents using *Drd1a*-ChR2 mouse evoked feed forward IPSCs in SNr GABA neurons that were blocked by Gbz (n = 5 cells, p<0.001, paired two tailed *t* test). Histograms of measured IPSCs latency (**D**), rise time (**E**) and decay tau (**F**) for feed forward and feedback inhibition revealed feedback inhibition has significantly faster kinetics compared with feed forward inhibition. (**G**) Average IPSC amplitude as a function of stimulus duration for feed forward and feedback inhibition. (**H**) Fraction of IPSC$_1$ amplitude during a 10 Hz train of photostimulation for feed forward and feedback inhibition. For **D**–**H**; maroon traces represent data from *Gad2*-ChR2 mice measuring feedback inhibition, n = 15 cells; blue traces represent data from *Drd1a*-cre mice measuring feed forward inhibition, n = 18 cells; for **D**–**H**, p<0.001, paired two tailed *t* test.
The following figure supplements are available for figure 5:

**Figure supplement 1**. Feedback inhibition provides fast, transient inhibition.

differences presumably reflect the non-homogeneous expression of ChR2 in the virally infected SNr of *Gad2*-ChR2 mice.

## Spatial organization of the local inhibitory microcircuit of the substantia nigra

Our results demonstrate a previously unappreciated potency of feedback inhibition in the SNr (***Figure 5G***). Collateral synapses provide sufficient inhibition to regulate the gain of the output of the

basal ganglia even during strong activation of the network (*Figure 3G*). However, anatomical reconstruction of individual axons suggests a sparse connectivity within the SNr (*Mailly et al., 2003*). The modest amplitude of individual mIPSCs (~150 pS; *Figure 3—figure supplement 2*) and the low cell density of the SNr (*Gerfen, 2004*) (~30,000 neurons in a ~4 mm³ volume [*Oorschot, 1996*]) imply that feedback inhibition derives from a substantial volume. The maximal amplitude of evoked IPSCs was 10,000 pS. If we assume a <1% connection probability then we would predict that inhibition would be derived from neurons in a ~600 μm radius from a given postsynaptic neuron. Consistent with such a model our wide-field stimulation experiments suggested that feedback inhibition magnitude scaled similarly for pairs of neurons (*Figure 3I*). While reconstructions of individual axons have been studied in detail, such results cannot be used to reliably infer the convergence of input onto an individual projection neuron. Moreover, light-level anatomy data cannot reliably predict the functional impact of feedback inhibition. Thus, we next adapted the ChR2-assisited circuit mapping (CRACM) (*Petreanu et al., 2007*; *Wang et al., 2007*) method developed for the neocortex to study the interconnectivity within nigral microcircuit.

Using a 10X objective, it was possible to contain the entire extent of the SN within a single field of view. We positioned either 81-point or 140-point grids of stimulation sites to cover the SN (*Figure 6A*). To achieve high spatial resolution of ChR2 activation, we used a focused 470 nm laser beam that could be rapidly re-positioned to each point on the grid in a pseudorandom sequence that avoided nearest neighbors. The duration of the light pulse was gated so as to deliver brief (<1 ms) pulses of light at each stimulation site. To obtain reliable and spatially homogeneous expression of ChR2 across nigral projection neurons, we used slices from the *Thy1*-ChR2 transgenic mouse line. Relative to wide-field stimulation or stimulation targeting axonal fibers, we reduced the maximal power of the laser using neutral density filters and performed calibration experiments to find intensities that would evoke precise, time-locked spikes in a small number of neurons with somatodendritic arbors surrounding the stimulation site (*Figure 6—figure supplement 1*). Our data were consistent with a requirement for propagating action potentials to elicit postsynaptic responses and we found no evidence in recorded neurons of direct axonal stimulation under these conditions (*Figure 6—figure supplement 1*). Furthermore, we focused on the rising phase and initial peak of IPSCs to bias our analysis towards transmission that resulted from highly reliable, low jitter spikes initiated at each stimulus site.

To measure the spatial organization of local inhibitory connectivity within the SNr, postsynaptic GABA neurons were clamped to $V_h$ = +20 mV in the presence of glutamate receptor antagonists. The peak amplitude of light-evoked IPSCs was determined for each stimulus position and response maps of IPSC amplitude as a function of stimulus position were generated (*Figure 6A–C*). While the viral-overexpression preparations do not provide homogeneous expression, we found that the length scale of the intranigral microcircuit estimated using the *Gad2*-ChR2 mouse was consistent with that from the *Thy1*-ChR2 mouse (*Figure 6C*). To characterize the spatial organization of input to individual neurons, we characterized the 'receptive field' of feedback inhibition as the center of mass (COM) and the 25% isoinhibition contour (ISO; *Figure 6D*). Finally, the majority of the somatodendritic arbor of each neuron was reconstructed from image stacks acquired on a two-photon microscope (*Figure 6D*) that allowed us to estimate the COM of the dendritic arbor.

## Feed forward and feedback inhibition have independent spatial organization

It has been proposed previously that collateral inhibition may be largely confined to topographic boundaries defined by feed forward input from the striatonigral pathway (*Mailly et al., 2003*). However, if feedback and feed forward input were organized in register, then feedback could not produce a signal proportional to the 'global' activation of the network. This would imply a collection of parallel channels each of which could exhibit strong feedback. By contrast, our recording data in vivo (*Figure 1*) suggested that feedback was proportional to the average activation across a large spatial extent of the SN. Thus, we next asked whether the feedback intranigral inhibition was organized in register with feed forward inhibition from the striatum.

If feedback inhibition were organized in register with feed forward inhibition, then we would predict that (1) the somatodendritic position of the postsynaptic neuron should predict the location of the inhibitory receptive field and (2) the spatial extent of inhibitory receptive fields should be matched to the topographic boundaries defined by feed forward inhibition. In contrast to the first prediction, we found that there was no correlation between the location of individual neurons and the source of the

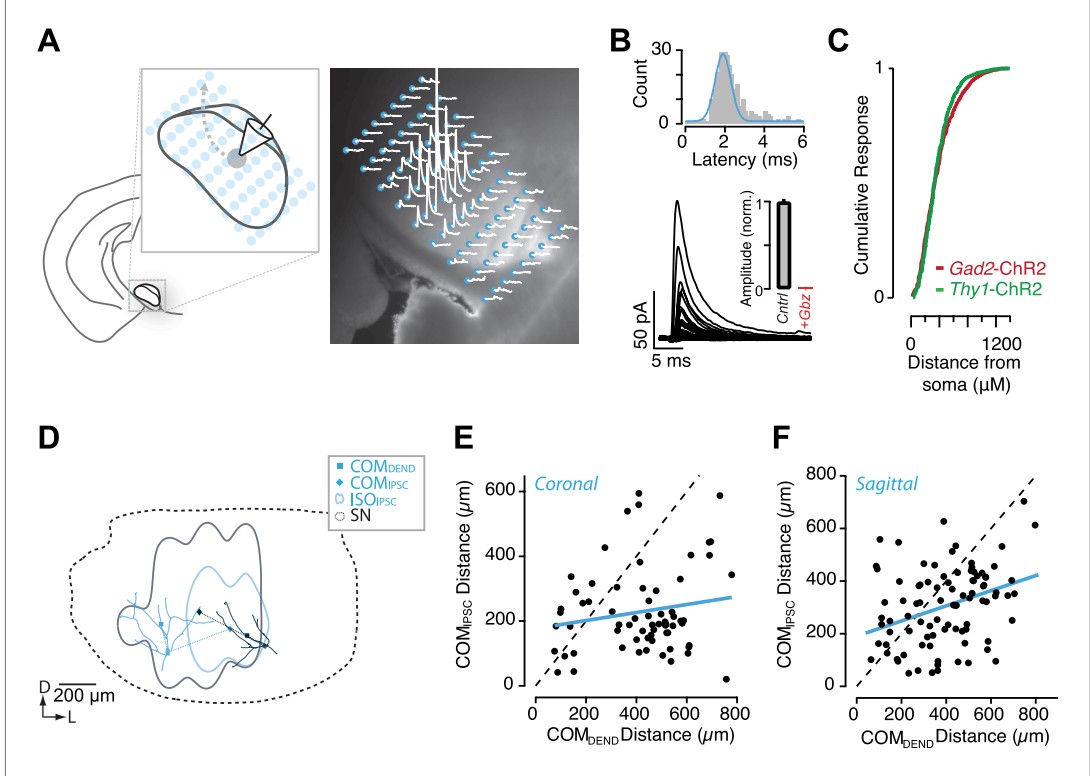

**Figure 6**. Circuit mapping of feedback inhibitory circuitry of SNr. (**A**, left) Schematic of the experimental configuration used for channelrhodopsin-assisted circuit mapping. Whole-cell voltage clamp recordings were obtained from SNr GABA neurons while a focus laser beam was scanned across the SNr to excite SNr neurons with high spatial resolution. (**A**, right) Postsynaptic responses to individual photostimulations (white) were aligned to the DIC image of the slice. Stimulation points are indicated by cyan. (**B**) Example of evoked IPSCs from a single recording with a histogram of IPSC latencies for all recordings. Evoked IPSCs were completely inhibited in the presence of Gbz (**B**, insert; n = 8 cells; p<0.001, paired two tailed *t* test). (**C**) Cumulative histogram of response magnitude as a function of the distance between the stimulation site and recorded neuron in the *Thy1*-ChR2 (green) and *Gad2*-ChR2 (maroon) preparations. (**D**) Example of IPSC maps for two neurons. The dendritic arbor of each recorded neuron was reconstructed and transformed into the common SN reference frame (dotted line). For each neuron the center of mass (COM) of inhibition (COM$_{IPSC}$, filled diamond), COM of dendritic field (COM$_{DEND}$, filled square) and the isocontour of 50% inhibition (IS0$_{IPSC}$, colored line) were calculated. (**E** and **F**) The COM$_{IPSC}$ was plotted as a function of the COM$_{DEND}$ for each neuron recorded in coronal (**E**; n = 14 cells) and sagittal (**F**; n = 16 cells) sections and the correlation fit estimated (blue line).

The following figure supplements are available for figure 6:

**Figure supplement 1**. Light-evoked IPSCs result from perisomatic spiking.

strongest inhibition within the SNr. In other words, we found that neurons with non-overlapping dendritic arbors could have largely overlapping inhibitory receptive fields (*Figure 6D*). We found no correlation between relative somatic position and the correlation of the inhibitory receptive fields in neither coronal (n = 14) nor sagittal (n = 16) slices (*Figure 6E–F*).

To assess whether local inhibition in the SNr observed topographic boundaries defined by afferent inhibition, we generated double transgenic mice in which ChR2 was expressed under control of the *Thy1* promoter and *cre*-recombinase was expressed under control of the D1 receptor (hereafter referred to as *Drd1a*-cre x *Thy1*-ChR2). We then made focal injections of a *cre*-dependent virus expressing a red fluorescent protein into the dorsal striatum 2–3 weeks prior to performing circuit mapping experiments in midbrain slices (*Figure 7—figure supplement 1*). Clear axonal labeling could be readily observed in the SNr (*Figure 7A*; *Figure 7—figure supplement 1*). The striatonigral projection exhibited the characteristic 'dual nature' that has been observed following focal tracer injections in

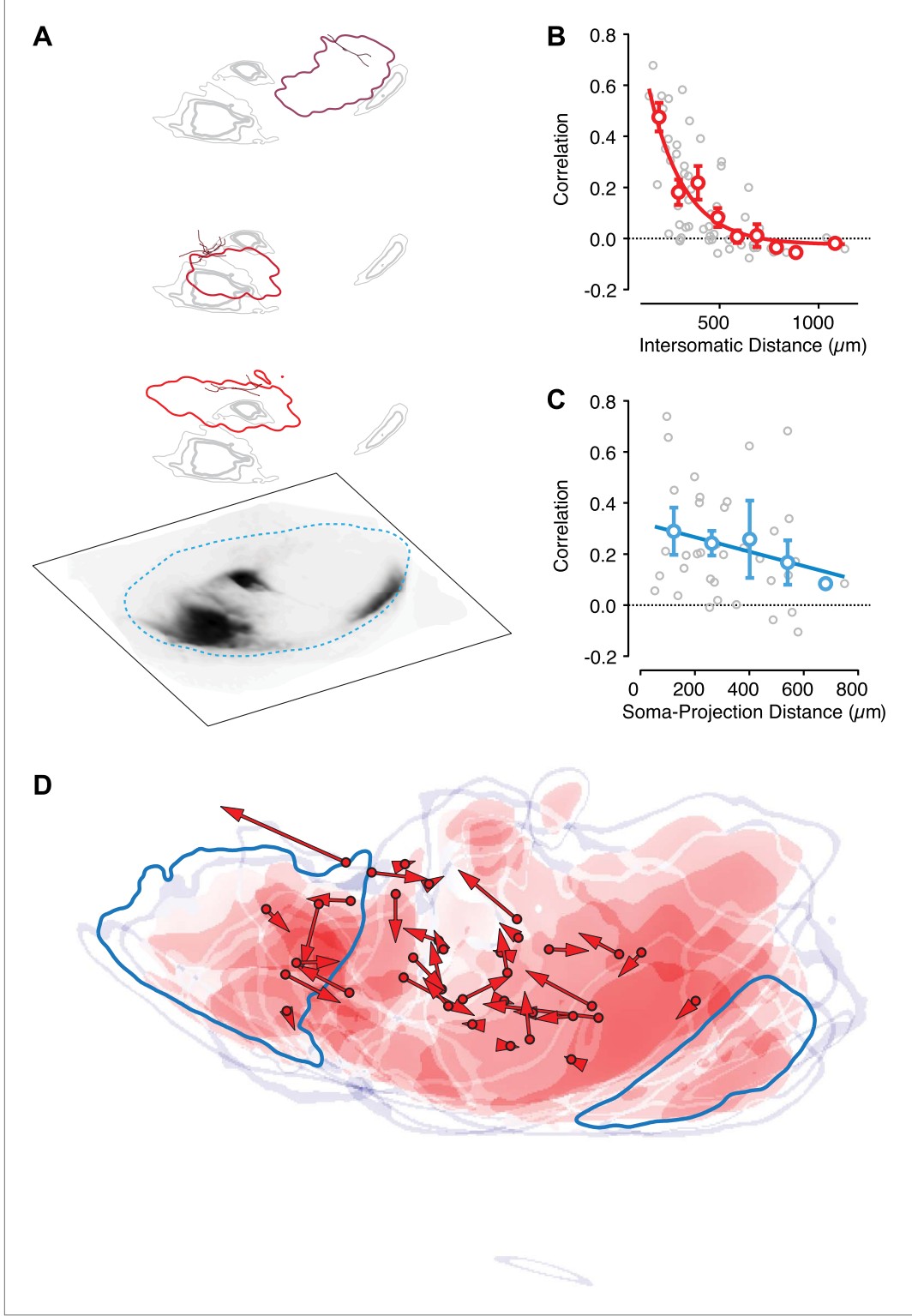

**Figure 7**. Intranigral inhibition is poorly predicted by the organization of the striatonigral pathway. Neurons in the dorsal striatum of *Drd1a*-cre x *Thy1*-ChR2 double transgenic mice were infected with cre-dependent AAV that drove the expression of a red fluorescent protein to label striatonigral axons (tdTomato). Bright-field images of the fluorescent axons in the SN were used to estimate the location of labeled axons (**A**, bottom layer). Estimates of the density of axonal labeling were produced by extracting the axon contour (quartiles indicated by gray line thickness) and compared with the localization of local inhibitory input (thresholded at 20% of maximum response) for multiple

*Figure 7. Continued on next page*

*Figure 7. Continued*

neurons recorded in the same slice (**A**, upper 3 layers). Individual postsynaptic neurons with proximal dendritic arbors reconstructed are shown in shades of red. The approximate border of the SN is indicated (cyan dashed line). (**B**) The correlation in spatial maps of IPSC amplitudes were computed for all pairwise comparisons between neurons recorded in the same slice (*n* = 10 slices; *n* = 36 cells) as a function of the distance between somata. Gray circles are individual correlations, red circles are binned means with standard errors, and solid red line is an exponential fit. (**C**) For each slice the correlation between a spatial map of IPSC amplitudes and the axonal density map is shown as a function of the distance between the soma of the recorded neuron and the center of mass of the axon projection. (**D**) For all slices the maximum intensity contrast ('Materials and methods') for the axonal labeling was overlaid with the location of all recorded somata (red circles). The angle and distance to the center of mass of the spatial maps of IPSC amplitudes are indicated by the red arrows. An example projection field from a single infection of the dorso-medial striatum is shown in dark cyan.

The following figure supplements are available for figure 7:

**Figure supplement 1**. Mapping striatonigral axonal terminal fields.

**Figure supplement 2**. Anatomical organization of the striatonigral pathway.

rats (*Gerfen, 2004*). In each slice, we performed circuit mapping for three to six projection neurons at a range of distances from the axonal termination fields (*Figure 7A*). We found that the correlation in the maps obtained from individual neurons was a monotonically decreasing function of distance between neurons (*Figure 7B*). However, we found no clear organization between the striatonigral projection and the maps of feedback inhibition (*Figure 7A,C*). The organization of the local inhibitory circuit in relation to the boundaries of striatonigral axonal tracing suggests that there could be a partial separation between the regions that receive input from the medial and lateral striatum (*Figure 7D*, *Figure 7—figure supplement 2*). However, the consistent fall off of the correlation in inhibitory response maps across neurons (*Figure 7B*) suggests that projection neurons receive feedback inhibition from a diffuse microcircuit and independent of the discontinuous topography of feedforward input.

## The spatiotemporal properties of intranigral inhibition in vivo

The potency and diffuse organization of feedback inhibition suggested that collateral synapses from SNr projection neurons could be a powerful determinant of the activity of the SNr even in the presence of ongoing input from afferent sources. The in vitro preparation presumably reduces or eliminates structured activity in afferent sources of input that could either directly compete with or modulate the consequence of feedback inhibition. Moreover, the length constant of inhibition was similar in the sagittal and coronal planes (*Figure 6E,F*) and thus, the potency of inhibition measured in vitro is, if anything, an underestimate of the impact of intranigral inhibition. We next asked whether we could use a complementary approach to measure the extent of the intranigral microcircuit in the awake mouse. By contrast to the mapping experiments above where we measured at one location (neuron) and stimulated many other locations, we next used either a silicon probe electrode array (*Figure 8A*) or wire array (*Figure 8—figure supplement 1*) with integrated optical fibers to stimulate at one (somewhat diffuse due to light scattering) location while measuring the spiking activity of SNr neurons at many neighboring locations.

Following implantation of a 64-site silicon probe array with integrated optical fiber into the SN of *Gad2*-ChR2 mice (*Figure 8A*), we observed single units with narrow waveforms and high baseline firing rates characteristic of SNr projection neurons (*Figure 8B*). The distribution of baseline firing rates obtained in vivo was closely matched to the same distribution obtained using on-cell recordings from identified SNr projection neurons in vitro (*Figure 8C*). We also confirmed that brief stimulation with a light pulse, like the stimulus used in the gain experiments in *Figure 3*, produced population responses similar in time course to those measured during response to natural stimuli (*Figure 3—figure supplement 1*). We found that in vivo, as observed in vitro, stimulus pulses of increasing duration produced mixed responses in the SNr population characterized by units with direct excitation and a subsequent delayed inhibition (*Figure 8B*, upper example) as well as units with a pure suppression of firing that onset with a short delay (~5 ms latency; *Figure 8B*, lower example).

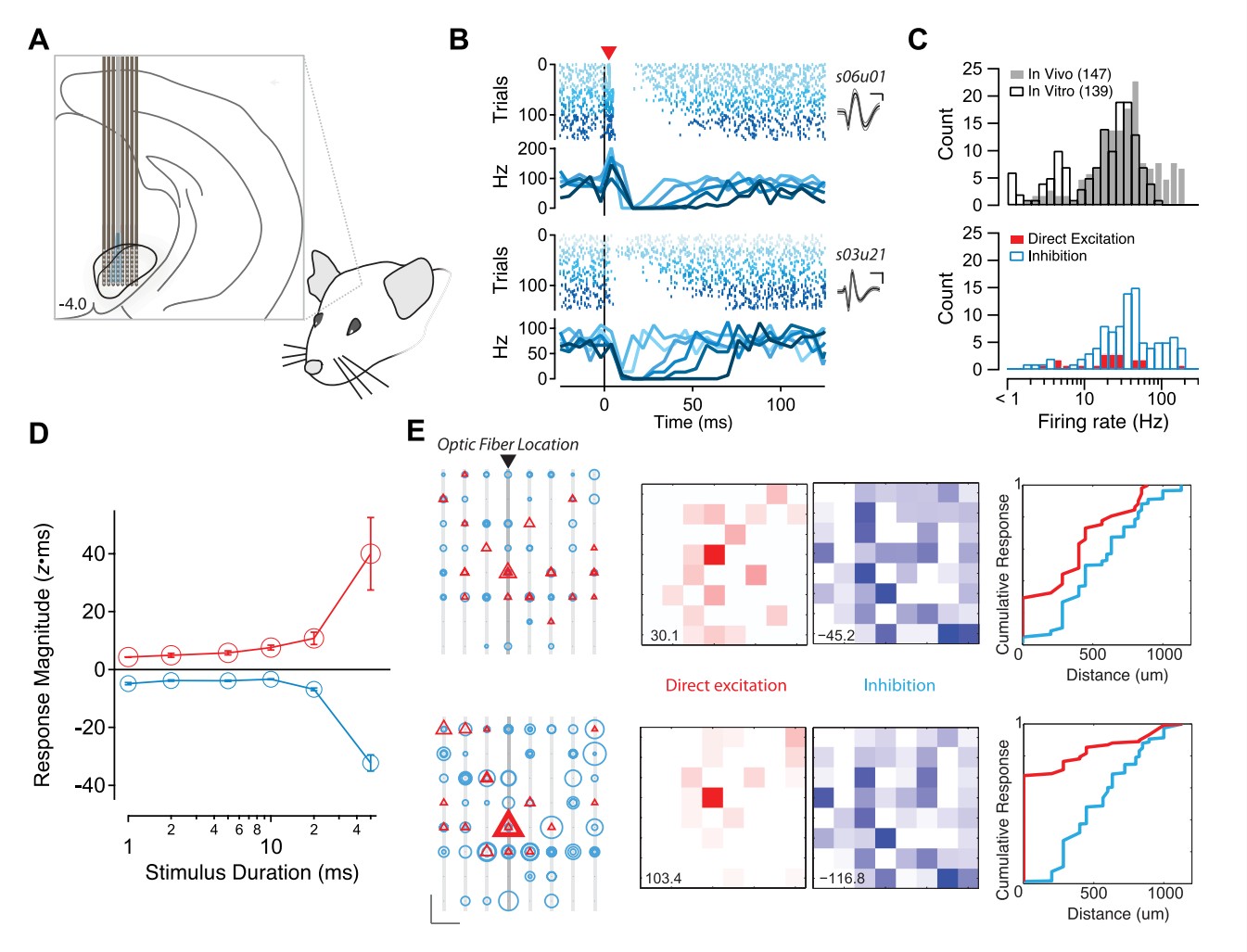

**Figure 8**. Potent and diffuse intranigral inhibition in vivo. (**A**) Schematic of experimental configuration used for in vivo recordings. An optical fiber was affixed to one shank of a silicon probe electrode array. The array was lowered into the SN of awake, head-fixed mice. (**B**) Raster plots of responses to light stimulation for two example single units isolated from such recordings. Spikes are indicated by vertical hash marks, colored and sorted by stimulus duration. Mean PSTHs are shown in lower panels and the average waveform (±1 SD) are shown in the insets. Some units (e.g., s06u01) exhibited direct excitation by photostimulation followed by suppression. While other units (e.g., s03u21) located at a more eccentric position on the array exhibited a delayed (one 5 ms PSTH bin) suppression of firing. (**C**) The distribution of average firing rates was very similar for single units isolated in vivo (gray bars) and on cell spiking rates observed in vitro (open bars). Directly excited units (red bars) and units exhibiting inhibition below baseline (open cyan bars) are plotted as a function of baseline firing rate. (**D**) The mean response magnitude for units exhibiting short latency activation (red) and the most inhibited quartile of the population (cyan) are plotted as a function of stimulus duration. (**E**) The spatial arrangement of sites at which direct excitation (red) or inhibition below baseline (cyan) was observed. Left, individual shanks of the silicon probe array are shown as light gray lines and the shank to which the optical fiber was affixed is shown in darker gray. The position of individual recording sites is represented as black dots. Every significant excitatory and/or inhibitory response is represented as a triangle or circle, respectively. Scale bar: 200 µm, 5 z×ms. The diameter of the symbol reflects the magnitude of the response to stimulation for stimuli of 20 (upper) and 50 (lower) ms stimuli. Middle, a maximum intensity projection for direct excitation (red) and inhibition (blue). Scaling of maximal value is shown in lower left. Right, a cumulative histogram of response magnitude as a function of distance from the focus of excitation (most strongly activated site on the shank with associated optical fiber). Distance calculated based upon 200 µm site spacing on the silicon probe array.

The following figure supplements are available for figure 8:

**Figure supplement 1**. Feedback inhibition shapes SNr output in Thy1-ChR2 transgenic mice.

A relatively constant level of inhibition combined with increasing excitatory drive can produce subtractive effects on the output spiking of a neuron. By contrast, divisive gain effects require that inhibition be recruited in proportion to changes in excitatory drive. Our in vitro data were consistent with a

divisive gain effect on nigral output due to intranigral inhibition. To distinguish these two possibilities in vivo, we examined neurons that exhibited apparent direct excitation (short-latency increase in stimulus evoked firing) with those that exhibited inhibition (stimulus evoked suppression of firing below baseline). We found that inhibition and excitation were recruited with a similar dependence on stimulus intensity consistent with a model in which intranigral inhibition produces a divisive gain effect on the output of SNr projection neurons (*Figure 8D*).

Finally, our circuit mapping data (*Figures 6 and 7*) suggested that functional inhibition extended for hundreds of microns within the SNr. To estimate the spatial extent of inhibition in vivo, we next examined the position of excitatory and inhibitory responses on the electrode array. The location of each inhibitory and excitatory response is plotted as a function of position for the 20 ms and 50 ms stimulus conditions as a function of electrode position, maximal intensity projection, or cumulative histogram as a function of distance from the focus of excitation (*Figure 8E*). Taken together, these data indicate that functional inhibition can extend for hundreds of microns beyond the focus of excitation in vivo as was observed in vitro.

## Discussion

The control of voluntary movement is well described by optimal feedback control models (*Wolpert and Ghahramani, 2000*). The basal ganglia is part of the extended brain circuit that controls voluntary movement (*Kandel et al., 2000*). Within this extended circuit there are a number of potential feedback projections (*Dudman and Gerfen, 2014*). However, the only intrinsic source of feedback to the main basal ganglia output nucleus, the SNr, is the microcircuit formed by collaterals of projection neurons (*Deniau et al., 2007b*). Despite anatomical (*Mailly et al., 2003*; *Deniau et al., 2007b*) and functional (*Tepper et al., 1995*, *2007*) evidence for the existence of this collateral microcircuit, its functional organization, relative impact, and properties were largely unknown. Here, we show that the unique properties of the SNr microcircuit—both intrinsic properties of projection neurons and the organization of functional activity—combine to implement a potent feedback inhibitory circuit that can exert a divisive gain control effect on the basal ganglia output.

In behaving mice, we observed that the SN microcircuit appears to modify transient responses to salient sensory stimuli, in our case a CS, in a manner consistent with negative feedback (*Figure 1*). We subsequently used cell-type specific expression of ChR2 to show that (1) the inhibition provided by collaterals of SNr projection neurons was sufficient to dramatically suppress firing of projection neurons even in the presence of strong activation (*Figure 3*); (2) collateral synapses produced inhibition with a rapid onset and modest short-term depression allowing for sustained inhibition during repetitive firing (*Figure 4—figure supplement 2*, *Figure 5*); (3) intranigral inhibition had distinct biophysical properties but comparable magnitude to the major source of feed forward input, the striatonigral pathway (*Figure 5*); (4) individual intranigral synapses were weak, but potent inhibition resulted from a spatially diffuse microcircuit in vitro (*Figures 6 and 7*) and in vivo (*Figure 8*). Together, these data indicate that the functional architecture of the intranigral microcircuit is sufficient to provide robust feedback inhibition that is proportional to the activity SN population—that is it provides a potent gain control on the output of the basal ganglia.

Finally, existing anatomical studies were divided on whether the striatonigral and intranigral circuits were organized in spatial register (*Grofova et al., 1982*; *Mailly et al., 2003*). The diffuse sources of collateral input to SNr projection neurons suggested that feedback inhibition extended across topographic divisions. We further confirmed this observation by combining axonal tracing together with circuit mapping to demonstrate that the striatonigral and intranigral projections were organized independently (*Figure 7*). Thus, our data imply that the gain control provided by the intranigral microcircuit could reflect a global negative feedback signal that spans individual channels of feed forward basal ganglia activity.

### Optogenetic-based measurements of the intranigral microcircuit

In our experiments, we used two approaches to achieve cell-type specific expression of ChR2 in the GABAergic projection neurons of the SNr. Both viral-mediated infection of GABAergic neurons in the SNr and the *Thy1*-ChR2 transgenic mouse line exhibited cell-type specific expression of ChR2 in GABAergic neurons of the SN, but not in dopamine neurons. GABAergic projection neurons and dopamine neurons are thought to be the only two cell types present in the rodent SN. Importantly, both approaches had indistinguishable properties of inhibition onto both GABAergic projection

neurons (*Figure 5—figure supplement 1*) and onto dopamine neurons (*Pan et al., 2013*). Moreover, viral mediated expression allowed for the best possible quantitative comparison between feed forward and feedback pathways by ensuring that expression of the ChR2 was controlled by a common promoter (*Figure 5*). Thus, for these results the only significant differences between the virally-mediated and transgenic expression of ChR2 in the SNr was the fraction of the population found to express ChR2 and the resultant magnitude of feedback inhibition. While viral-mediated expression could be used to estimate the divergence of inhibitory projections from a single stimulation site (*Figure 8*), it was not suitable for comprehensive mapping of convergence onto individual SNr neurons (*Figures 6 and 7*). The concern in a transgenic mouse line is that ChR2 expression could be present in afferent fibers. To control for this, we blocked excitatory transmission, confirmed that light-activated currents had kinetics expected of ChR2 positive neurons, and confirmed that neurons of the major source of inhibitory input, the striatonigral pathway, did not express ChR2. Consistent with these observations it has previously been suggested that inhibitory fibers in this *Thy1*-ChR2 mouse line do not express ChR2 (*Gradinaru et al., 2009*). However, in addition to expression in the SNr, we did also observe that neurons of the external globus pallidus (GPe) express ChR2. The GPe is the source of a projection to the subthalamic nucleus and a more modest projection to the SNr (*Bolam et al., 2000*; *Gerfen, 2004*; ). Nonetheless, we found that potent feedback inhibition could be observed in *Gad2*-ChR2 mice and that stimulation of GPe axons yielded relatively less inhibition in SNr neurons than either the striatonigral or intranigral pathways under our stimulus conditions. Finally, we used laser powers that were attenuated relative to the powers necessary to directly stimulate severed axons in these mice (data available from the authors on request), consistent with our observation that only perisomatic stimulation was sufficient to evoke reliable spiking in SNr neurons (*Figure 6—figure supplement 1*).

## A mechanism for divisive gain control in a circuit lacking interneurons

Unlike many other circuits in which gain control has been studied, projection neurons of the SNr are spontaneously active. Divisive gain control requires that there is little effect of inhibition in the absence of stimulation. In the case of gain control mediated by an interneuron this can be achieved in a number of ways, for example, through facilitating inhibitory synaptic transmission (*Silver, 2010*; *Figure 9*). However, in spontaneously active neurons, it is less clear how to prevent feedback inhibition from altering the baseline firing rate as we observe here. One possibility suggested by our data is that a broad distribution of firing rates (*Figure 8*) combined with relatively weak individual connections (*Figure 4*) could produce inhibition that is essentially tonic and too small to significantly affect intrinsic currents that drive repetitive spiking (*Figure 4—figure supplement 1*) and so is counteracted via sub-threshold inward currents necessary for repetitive firing. We provided support for such a model by demonstrating that there is, indeed, a high frequency (200 Hz) of spontaneous IPSCs bombarding SNr projection neurons (*Figure 4*). Further consistent with this model, simulation of a steady back-ground rate of IPSCs using dynamic clamp revealed a relative insensitivity of spike rate to a tonic net inhibitory background of inputs (*Figure 4*). This balance in the tonic firing rate can be disrupted by synchronously recruiting neighboring neurons to produce a rapid inhibition that, combined with the positive feedback produced by inhibition and subsequent disinhibition, overcomes via de-activation (*Nolan et al., 2003*) the inward currents that drive the membrane potential towards threshold. This mechanism combining intrinsic properties and synaptic properties is sufficient for a population of spontaneously active inhibitory neurons to implement divisive gain control in the apparent absence of interneurons (*Deniau et al., 2007b*). To our knowledge this represents a novel circuit mechanism for divisive gain control (*Silver, 2010*; *Figure 9*).

## The multiple roles of intranigral inhibition: feedback vs lateral inhibition

One can think of two possible regimes in which the SNr may operate and each has distinct implications for the function and role of intranigral inhibition. On the one hand, it has been argued for some time that the feed forward pathways of the basal ganglia are topographically organized and largely independent (*Mink, 1996*; *Haber, 2003*; *Hikosaka, 2007*). Movement is thus thought to occur when a focal population of projection neurons becomes inhibited by feed forward input and thus disinhibits downstream pre-motor structures. From this perspective, the diffuse intranigral microcircuit could act to release neighboring projection neurons from intranigral inhibition and thereby suppress unintended movements. Thus, during focal activation of the SNr, collateral inhibition may be thought of as

a mechanism for contrast enhancement akin to the role of lateral inhibition in sensory systems. This mechanism may be reflected in the time locked, bidirectional changes in the firing of SNr projection neurons that are commonly observed prior to and during movement in mice (*Pan et al., 2013*; *Fan et al., 2012*) and primates (*Turner and Anderson, 1997*; *Nevet et al., 2007*).

On the other hand, while the direct striatonigral pathway is topographically organized there is considerable divergence in the corticostriatal input in mice (*Pan et al., 2010*). In addition, there may be less precise topographic organization of the indirect pathway that enters the SNr via the subthalamic nucleus (*Bolam et al., 2000*). The subthalamic nucleus also receives direct cortical input and ascending input from the midbrain (*Coizet et al., 2009*) and hindbrain (*Bevan and Bolam, 1995*; *Winn, 2006*). The topographic organization of these ascending pathways is less well understood. Regardless of the topographic precision, the inputs that arrive at the SNr from the subthalamic nucleus convey a great diversity of information and likely exhibit a diversity of dynamics. While there are mechanisms that could maintain activity within a fixed dynamic range in upstream structures (*Silver, 2010*), individual neurons that constitute the output of cortical areas project to multiple subcortical structures (*Kita and Kita, 2012*). It is therefore unlikely that the dynamic range of even the cortical output is appropriate for the diverse computations performed in all target structures. To effectively control the basal ganglia output in the presence of such diverse input dynamics and anatomical divergence would seem to require coordinated processing across functional domains. Thus, divisive gain control supplied by a diffuse but potent inhibitory microcircuit could be well suited to ensure that activity remains within a fixed dynamic range.

We favor a model in which these contrasting descriptions of the role of the intranigral microcircuit are two aspects of its function that can be engaged in different input regimes. The diffuse organization

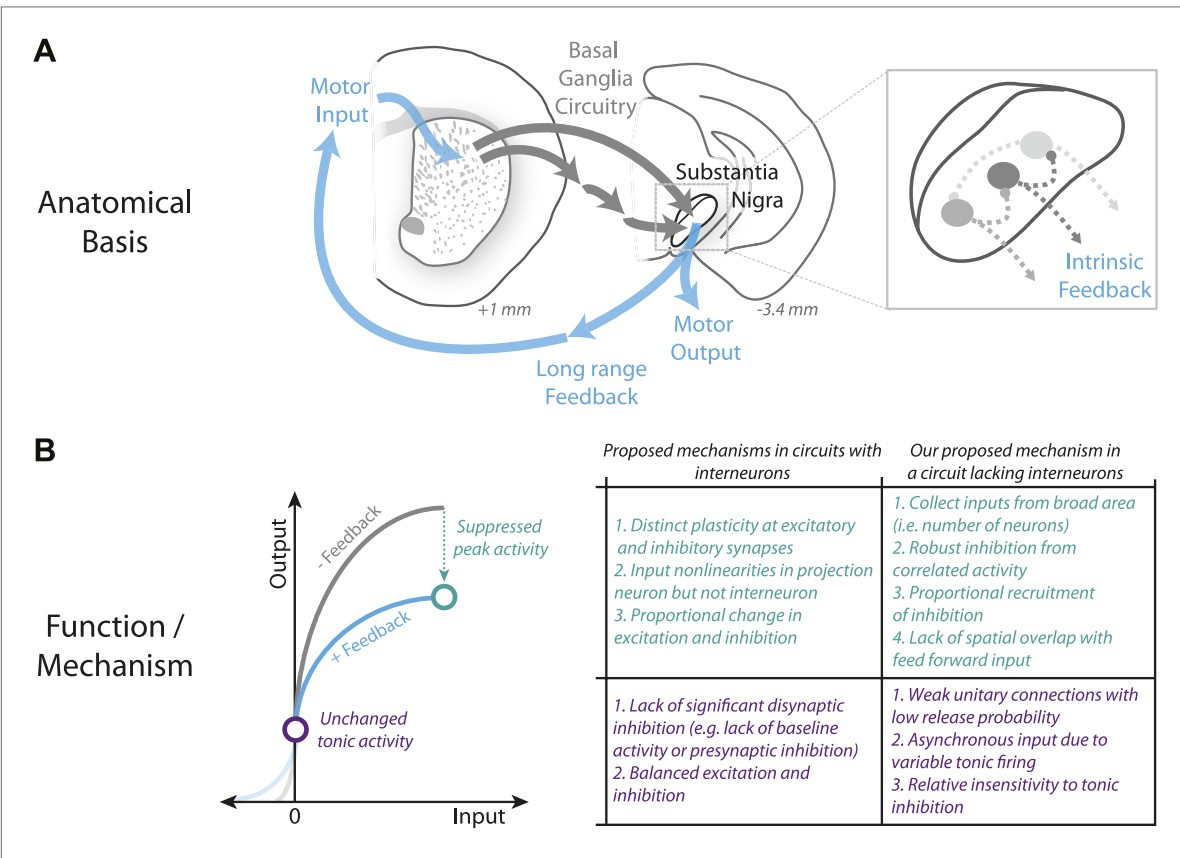

**Figure 9**. Schematic summary of proposed mechanism for divisive gain control in a circuit lacking interneurons. (**A**) Schematic of the canonical basal ganglia circuit with detail showing the anatomical basis for intrinsic feedback control of the basal ganglia output via the intrinsic microcircuitry of the substantia nigra. (**B**) Comparison of candidate mechanisms for gain control described in microcircuits with interneurons (e.g., *Silver, 2010*) with the mechanism for divisive gain control in the substantia nigra (a circuit thought to lack interneurons) described here.

that could produce lateral inhibition in some regimes under the animal's control (i.e., activation of specific well learnt actions), but, may be necessary to produce divisive gain control in other regimes (i.e., 'global' activation of the SNr by salient stimuli). While it is not currently possible to monitor nigral dynamics during selective manipulation of feedback, but not feed forward or efferent inhibition, such an approach will be required to definitively test the predictions of our work.

### The impact of intranigral inhibition on the function of the basal ganglia

Simple geometrical considerations suggest that the intranigral microcircuit integrates functionally distinct information on a large scale. Cortical afferents to the basal ganglia are derived from an estimated 17 million neurons (*Zheng and Wilson, 2002*) spanning the majority of the roughly 100 mm³ volume of neocortex. These inputs are funneled through the basal ganglia and will ultimately terminate on approximately 30,000 projection neurons (*Oorschot, 1996*) within the roughly 4 mm² volume of the SN. Dendrites of nigral projection neurons and intranigral inhibition that extends over hundreds of microns (*Grofova et al., 1982*; *Mailly et al., 2001*) could therefore shape activity derived from cortical inputs separated by several millimeters.

Existing functional models focus on the feed forward structure of processing within the intrinsic basal ganglia circuitry (e.g., *Albin et al., 1989*; *DeLong, 2000*; *Mink, 1996*). However, our observation that intranigral inhibition is strong relative to the major source of feed forward input suggests that local processing of diverse streams of information in the SN could be critical for generating dynamics in the basal ganglia output. We hypothesize that the diverse impairments characteristic of pathological disruption of basal ganglia function could reflect, in part, a control system operating outside of a stable regime. Although there is evidence of perturbed dynamics in the SNr in disease models (*Ibanez-Sandoval et al., 2007*; *Samadi et al., 2008*; *Wang et al., 2010*), the specific contribution of the intranigral microcircuit to the diverse behavioral impairments observed in diseases afflicting the basal ganglia circuit remains unclear.

## Materials and methods

### Animals

For in vitro experiments, adult transgenic mice (10–30 weeks old) expressing either; ChR2-YFP fusion gene under the control of the mouse thymus cell antigen 1 promoter (Line 18, Stock #007612; Jackson Labs, Bar Harbor, Maine, '*Thy1*' mice), *cre*-recombinase under the control of the glutamic acid decarboxylase 2 gene (Stock #010802; Jackson Labs, 'GADcre' mice) or *cre*-recombinase under the control of the dopamine receptor D1A (GENSAT, *Gong et al., 2010*, Rockefeller University, '*Drd1a*-cre' mice). For in vivo electrophysiology experiments, four adult (30 g, 3–6 months old) *Thy1* mice were used. All animals were handled in accordance with guidelines approved by the Institutional Animal Care and Use Committee of Janelia Farm Research Campus. The experimenter was not blinded to genotype.

### Animal care

Mice were housed in a temperature- and humidity-controlled room maintained on a reversed 12 hr light/dark cycle. For in vivo physiology experiments, mice were housed individually. For in vitro experiments, mice were group housed (1–5 mice per cage).

### Viral expression

We used 3 adeno-associated viruses (AAV, serotype 2/1) to achieve either conditional expression of ChR2 and tdTomato or pan-neuronal expression of eGFP and tdTomato. Viruses were produced at the Molecular Biology Shared resource of Janelia Farm Research Campus. Where indicated similar viruses can be obtained publicly from the Gene Therapy Program at the University of Pennsylvania (http://www.med.upenn.edu/gtp/vectorcore/Catalogue.shtml). Conditional ChR2 expression was achieved with AAV2/1 SYN-FLEX-ChR2-GFP *analogous to* AV-1-18917P. Conditional tdTomato expression was achieved with AAV2/1 CAG-FLEX-tdTomato-WPRE-bGH *available as* AV-1-ALL864. Pan-neuronal expression of tdTomato and eGFP were obtained via AAV2/1 SYN-[tdTomato/eGFP]-WPRE-SV40 *available as* AV-1-PV1696.

Viruses were injected into the striatum (STR) of *Drd1a*-cre mice, globus pallidus (GP) or substantia nigra (SN) of GADcre mice, in a fashion similar to that previously described (*Atasoy et al., 2008*). Briefly, under deep anesthesia, a small craniotomy was made over the STR (0.5 mm anterior-posterior, 1–2 mm medial-lateral, −2.5 mm dorso-ventral), GP (−0.45 mm anterior-posterior, −1.8 mm medial-lateral, −3.6 mm dorsal-ventral) or SN (−3 mm anterior-posterior, 1 mm medial-lateral,

−4.2 mm dorso-ventral). A glass pipette was used to pressure inject small volumes of virus (20–100 nl per injection site). Animals were allowed to recover for at least 2 weeks following surgery.

## In vitro electrophysiology

Briefly, adult mice were deeply anaesthetized under isoflurane, decapitated, and the brains were dissected out into ice-cold modified artificial cerebral spinal fluid (aCSF) (52.5 mM NaCl, 100 mM sucrose, 26 mM $NaHCO_3$, 25 mM glucose, 2.5 mM KCl, 1.25 mM $NaH_2PO_4$, 1 mM $CaCl_2$, 5 mM $MgCl_2$, and 100 µM kynurenic acid) that had been saturated with 95% $O_2$/5% $CO_2$. 300 µM thick coronal and sagittal slices (as indicated in the text) were cut (Leica VT1200S; Leica Microsystems, Germany), transferred to a holding chamber and incubated at 35°C for 30 min in modified aCSF (119 mM NaCl, 25 mM $NaHCO_3$, 28 mM glucose, 2.5 mM KCl, 1.25 mM $NaH_2PO_4$, 1.4 mM $CaCl_2$, 1 mM $MgCl_2$, 3 mM sodium pyruvate, 400 µM ascorbate, and 100 µM kynurenic acid, saturated with 95% $O_2$/5% $CO_2$) and then stored at 21°C.

For recordings, slices were transferred to a recordings chamber perfused with modified aCSF (119 mM NaCl, 25 mM $NaHCO_3$, 11 mM glucose, 2.5 mM KCl, 1.25 mM $NaH_2PO_4$, 1.4 mM $CaCl_2$, 1 mM $MgCl_2$, 3 mM sodium pyruvate, 400 µM ascorbate, saturated with 95% $O_2$/5% $CO_2$) and maintained at 32–34°C at a flow rate of 2–3 ml min$^{-1}$. For mIPSC experiments, extracellular $Ca^{2+}$ was replaced with 2 mM $Sr^{2+}$ to desynchronize release. Patch pipettes (resistance = 5–8 MΩ) were pulled on a laser micropipette puller (Model P-2000; Sutter Instrument) and filled with one of the following intracellular solutions: Current-clamp recordings of spike activity used a potassium gluconate-based intracellular solution (137.5 mM potassium gluconate, 2.5 mM KCl, 10 mM HEPES, 4 mM NaCl, 0.3 mM GTP, 4 mM ATP, 10 mM phosphocreatine, pH 7.5). Voltage-clamp recordings for IPSC measurements used a CeMeSO$_4$-based intracellular solution (114 mM CeMeSO$_4$, 4 mM NaCl, 10 mM HEPES, 5 mM QX314.Cl, 0.3 mM GTP, 4 mM ATP, 10 mM phosphocreatine, pH 7.5). Alexa Fluor 488 or Alexa Fluor 568 was commonly added to intracellular solution to aid cell visualization and post hoc reconstruction. In some experiments the following were added as indicated in the text: 10 µM CNQX or 5 µM NBQX, 50 µM D-AP5, 10 µM gabazine (Gbz), 0.5 µM tetrodotoxin (TTX). All drugs were obtained from Tocris Biosciences. Intracellular recordings were made using a MultiClamp700B amplifier (Molecular Devices) interfaced to a computer using an analog to digital converter (PCI-6259; National Instruments) controlled by custom written scripts (to be made available at http://dudmanlab.org/) in Igor Pro (Wavemetrics). Photostimulation was carried out using a dual scan head raster scanning confocal microscope and control software developed by Prairie Systems, and incorporated into a BX51 upright microscope (Olympus America).

Individual neurons were patched under DIC optics with a water-immersion 40X objective. Spiking was measured in the cell-attached configuration. The spiking frequency and action potential waveform were used to classify neurons as DA or GABA as described previously (**Pan et al., 2013**). Upon break in while diffusion of QX314 was allowed time to progress, negative voltage clamp steps were used to measure the hyperpolarization-activated inward (Ih) current. The presence of detectable inward currents was diagnostic for DA neurons. In current clamp recordings lacking QX314 in the internal solution the intracellular spike waveform and spontaneous firing frequency were further used to confirm cell identity. Analysis of postsynaptic currents (direct photocurrents, sIPSC, mIPSCs and evoke IPSCs) and spiking was performed using custom written analysis code in Igor Pro (Wavemetrics). Peak current amplitude was measured as the peak synaptic current relative to the baseline holding current preceding each stimulus. Tonic current amplitude was measured as the peak evoked synaptic current relative to the holding current preceding each train. The conductance (g) underlying IPSCs were calculated from $g = IPSC_{peak}/(V_m − E_{Cl})$, where $IPSC_{peak}$ is the peak amplitude of the IPSC and $V_m$ is the holding voltage. The equilibrium potential of GABA$_A$ current ($E_{Cl}$) was estimated at −70 mV from the Nernst equation. Rise time constants of postsynaptic currents were measured by finding the 20–80% slope of the rising phase of the stimulus-evoked current. Decay time constant of postsynaptic currents were measured by fitting a single exponential to the decay phase of the stimulus-evoked currents.

Spikes were detected at the threshold of maximum acceleration. Phase plots were constructed by plotting the first derivative of the somatic membrane potential (dV/dt) vs the somatic membrane potential for the average spike waveform. The membrane potential at which phase plot slope reached 10 mV·ms$^{-1}$ was denoted the voltage threshold and a linear fit was used to calculate the slope. The perithreshold slope was calculated as the slope of the 'kink' defined as the slope of dV/dt for 7 ms after the peak of the perithreshold dV/dt.

For dynamic clamp experiments, individual postsynaptic conductances were generated using IGOR Pro (Wavemetrics), from the sum of two exponentials with rise tau and decay tau derived from measured IPSC and EPSC rise and decay kinetics (rise tau = 0.5 ms, decay tau = 5 ms). The times of individual events were computed by sampling a Poisson distribution in which the rate of IPSC and EPSC events were independently changed from 1000 to 5000 Hz to generate different balances of excitation and inhibition. The convolved waveforms for excitation and inhibition were computed independently and passed to a custom made, digital dynamic clamp (update rate 30 kHz; to be described elsewhere) assuming reversal potentials of −70 mV and 0 mV for inhibition and excitation, respectively.

## Optical stimulation and imaging

The optics were designed to minimize the spread of the laser in the x, y dimensions of the focal plane while accentuating the focus in z by underfilling the back aperture of the objective. Stimulation intensity was controlled by pulse duration (0.2–1 ms). Stimulation typically consisted of 9 × 9 and 10 × 14 maps of stimulation sites with independent stimuli being delivered in a pseudo-random (non-neighbor) sequence at an interstimulus interval of ≥150 ms and values reflect the average of 3–4 repetitions of the mapping experiment for each cell. Stimulation strength was modulated by gating the laser at maximal power (473 nm, AixiZ or 488 nm, BlueSky Research) with varying durations using timing signals from an external pulse controller (PrairieView software) and the internal power modulation circuitry of the laser or an external Pockels cell (Conoptics) with indistinguishable results. Wide-field activation of ChR2 was accomplished using blue LED (470 nm, ThorLabs) transmitted through the fluorescence light path of the BX51 microscope. LED intensity and timing were controlled through a variable current source (ThorLabs). Stimulus families (input/output curves) were delivered in a pseudo-random order and repeated 3–10 times per cell.

## Analysis of mapping data

Analysis was performed using custom written routines for Igor Pro (Wavemetrics) and Matlab R2011a (Math Works). Analysis of the full field photostimulation was performed using standard analysis metrics as described in the text. To attempt to minimize the variability in estimates of short-term plasticity stimulation was performed at the half-maximum stimulus intensity determined by generation of an input–output function at the beginning of the experiment. The analysis of circuit mapping experiments was more complicated and is described briefly below and demonstrated more explicitly in *Figure 6*. Briefly, averages of 3–10 multisite photostimulation experiments were used in all analyses. The moment of photostimulation was determined by thresholding a photodiode signal positioned in a parallel light path to the stimulation light path. Galvonometer position signals were recovered from the PrairieView software and aligned using a transmitted light laser scanning DIC image of the brain slice. Offline analysis routines automatically detected the orientation of the stimulus grid and applied a rotation to put that grid into 0 rotation orientation. The dorsal and medial edges of the grid were manually annotated and used to flip or further rotate all grids into a common reference frame. The average dimensions of the SN are 1.67 mm wide by 1 mm tall. We found that the average of all grids had an identical (1.67–1) aspect ratio. There was modest variation (~10%) in the dimensions of the SNr grids. All grids were linearly stretched or compacted to the same mean aspect ratio and cell positions moved accordingly. Qualitatively similar results were obtained in the presence and absence of warping. Full depth maps were generated by convolving the response amplitude at individual stimulus positions with an empirically-estimated Gaussian response function. An isocontour of the resulting image was generated at the half maximum level using the 'contour' function supplied by Matlab. The center of mass (COM) was also calculated as the vector average of the Euclidean distance to the stimulus position weighted by response magnitude.

Reconstructions of recorded neurons were derived from two-photon fluorescent image stacks using the semi-automated software generously provided by Ting Zhao (Janelia Farm Research Campus, HHMI) (*Zhao et al., 2011*). The COM of the dendrites was calculated as the vector average with the weights defined by the width of the dendritic branch segment at the end of the vector position. Data were then loaded into Matlab for display and scaling.

## In vivo electrophysiology

Recordings were performed using either a 32-microwire arrays (CD Neural Technologies) or a 64-channel silicon probe array (NeuroNexus Technologies). Electrode arrays were stereotaxically

implanted under anesthesia (isoflurane; 1.5–2.5% in O$_2$) in mice that had been previously fitted with a plastic head restraint and held in place by a custom head fixation system (*Osborne and Dudman, 2014*). Design files and details about the manufacture and use of our head restraint system are available online (http://dudmanlab.org/html/rivets.html). Electrode arrays were targeted to the SN of the ventral midbrain (3.0–4.5 mm posterior to bregma, 0.5–2.0 mm lateral to midline and >3.5 mm below the surface of skull). Electrode arrays were maintained in position by a micromanipulator (Sutter Instruments or Scientifica) and connected to the recording systems via a flexible wire coupling and connector. For optogenetic experiments, a 200 µm core multimode fiber (ThorLabs) was affixed near the central recording wires of a 32 channel array or to one shank of the silicon probe array as indicated in *Figure 8A*. The entire array was slowly lowered in to the midbrain. Following >1 hr of recovery single unit recordings were obtained from alert, but quietly resting mice. Single cell isolation was performed offline using Offline Sorter (Plexon Technologies) and standard techniques. Analysis of stimulus-evoked responses were calculated and presented using Matlab 2011a.

The spike data in *Figure 1* are a subset of recording sessions (all sessions with ≥8 simultaneously recorded, putative GABAergic cells) from mice performing an auditory trace conditioning task described in detail previously (*Pan et al., 2013*). Briefly, mice were trained to consume sweetened water rewards delivered from a port placed on one wall of a behavior box. A speaker placed behind one wall of the box delivered pure tones (10 kHz; 500 ms duration) as conditioned stimuli (CS). Water rewards were delivered 2.5 s following CS onset. This data set included 599 single units recorded across sessions in which 5 to 21 units were recorded simultaneously. For each session, we computed the firing rate of the population of units prior to the onset of the CS and the transient response in the 200 ms following CS onset and subtracted the mean response across all trials. For each trial and all units recorded in a given session, we then computed the population response excepting the ith unit (PRE$_{population}$) and the response of the ith unit to the CS (RESP$_{single}$). For individual sessions, we determined the correlation between PRE$_{population}$ and RESP$_{single}$ for all units in the session. Significance of the correlation was determined using a permutation test. For the entire population, we plotted PRE$_{population}$ vs RESP$_{single}$ for all units, all trials. The data were binned into 20 equally spaced bins and mean data for all bins with more than five samples was plotted and fit with a sigmoid function using Igor Pro.

## Statistics

All statistical tests were performed using the statistics package from Matlab 2011a (Math Works). Paired comparisons were performed using the student's *t* test (all results were also confirmed with a non-parametric ranksum test). Multiple comparisons were performed using ANOVA. Significance was defined as $p < 0.05$ unless otherwise indicated. Averaged data are presented as mean ± standard error of the mean (SEM), unless otherwise specified.

## Acknowledgements

Susan Jones, Vivek Jayaraman, Alla Karpova, Albert Lee, Jeff Magee, Gabe Murphy, and Nelson Spruston provided critical feedback at various stages of preparation of the manuscript and progression of the project. We are indebted to the extensive feedback from our colleagues following presentation of this work at internal seminars on the Janelia Farm Research Campus.

JB is a graduate scholar in the Cambridge-Janelia Farm Graduate Program. JTD is a JFRC Group Leader of the Howard Hughes Medical Institute. This work was supported by funding from the Howard Hughes Medical Institute. The authors declare no competing financial interests. Correspondence and requests for materials should be addressed to dudmanj@janelia.hhmi.org.

## Additional information

### Funding

| Funder | Author |
| --- | --- |
| Howard Hughes Medical Institute (HHMI) | Joshua Tate Dudman |

The funder had no role in study design, data collection and interpretation, or the decision to submit the work for publication.

## Author contributions

JB, JTD, Conception and design, Acquisition of data, Analysis and interpretation of data, Drafting or revising the article; W-XP, Acquisition of data, Drafting or revising the article

## Author ORCIDs

Joshua Tate Dudman, http://orcid.org/0000-0002-4436-1057

## Ethics

Animal experimentation: This study was performed in strict accordance with the recommendations in the Guide for the Care and Use of Laboratory Animals of the National Institutes of Health. All of the animals were handled according to approved institutional animal care and use committee (IACUC) protocols (#08-36, 11-69) of the Janelia Farm Research Campus. The animal care and use program at Janelia Farm Research Campus is accredited by The Association for Assessment and Accreditation of Laboratory Animal Care, International (AAALACi).

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
