## [Decision Letter]

Thank you for sending your work entitled “The inhibitory microcircuit of the substantia nigra provides feedback gain control of the basal ganglia output” for consideration at e*Life*. Your article has been favorably evaluated by a Senior editor and 3 reviewers, one of whom is a member of our Board of Reviewing Editors.

The following individuals responsible for the peer review of your submission have agreed to reveal their identity: Reviewing editor: Sacha Nelson; peer reviewer: James Tepper.

The Reviewing editor and the other reviewers discussed their comments before we reached this decision, and the Reviewing editor has assembled the following comments to help you prepare a revised submission.

All three reviewers agreed that this is a thorough and carefully carried out study of feedback inhibition in the SNr, a topic of significant interest for understanding basal ganglia function.

One main point that could use further clarification concerns the distinction between subtractive versus divisive inhibition. This distinction does not seem central to the conceptual importance of feedback inhibition. How it behaves in vivo would seem to depend critically on the precise spatial pattern of the excitation driving it – i.e., whether it functioned as lateral inhibition (favoring one output over another), or spatially homogenous divisive inhibition (turning down all outputs) or even in a disinhibitory mode (removing inhibition by inhibiting the firing of neurons providing feedback inhibition). The fact that in operates divisively in the experiments shown in Figure 3 seems to reflect two factors: one that a wide field stimulus was used (e.g., as opposed to a smaller moving spot, or some other pattern) and second that there is no significant effect on baseline firing. This last point is somewhat counterintuitive. If individual SNr neurons are pooling feedback inhibition over a large pool (e.g., 70-100 neurons or more, spread out over a large spatial extent) and if each is firing spontaneously at 10-40 Hz, the tonic current is comprised of events at an aggregate rate of 700-4000 Hz. Even if the individual events were comparable to miniature events, this would seem to be a very large tonic current.

To clarify this, it would be straightforward to block spiking and measure the tonic current by washing in Gabazine. One could then compare the effects on spontaneous spiking of injecting a comparable negative current somatically in the absence of inhibition, or even inject a conductance if one wanted to get fancy. Potential concern about the electrotonic location of the feedback inhibition is mitigated by the fact that the events observed are rapid.

Although this proposed experiment might clarify the issue, it should be viewed as a suggestion and it may be feasible to clarify the issue with additional textual changes. Alternatively, it was also suggested that the manuscript would be improved by the inclusion of cell filling (at a minimum from slices but much better from in vivo staining) showing the axonal arborizations of the SNr projection neurons (this could also potentially address the concern above). If is not so, as the authors claim that the level of convergence of the SNr collaterals onto neighboring SNr (or DA) neurons cannot be reliably inferred from anatomical studies such as this, it would be more correct to say that this is beyond the scope of the present study, since many such studies have been performed in the laboratories of Somogyi, Bolam, Magill and others in the basal ganglia and elsewhere.

Finally, one reviewer felt that prior work supporting the idea that interneurons are absent in this structure should also be mentioned in the Discussion and not only raised in the Abstract.

---

## [Author Response]

*One main point that could use further clarification concerns the distinction between subtractive versus divisive inhibition. This distinction does not seem central to the conceptual importance of feedback inhibition. How it behaves in vivo would seem to depend critically on the precise spatial pattern of the excitation driving it – i.e., whether it functioned as lateral inhibition (favoring one output over another), or spatially homogenous divisive inhibition (turning down all outputs) or even in a disinhibitory mode (removing inhibition by inhibiting the firing of neurons providing feedback inhibition). The fact that in operates divisively in the experiments shown in*
Figure 3
*seems to reflect two factors: one that a wide field stimulus was used (e.g., as opposed to a smaller moving spot, or some other pattern) and second that there is no significant effect on baseline firing. This last point is somewhat counterintuitive. If individual SNr neurons are pooling feedback inhibition over a large pool (e.g., 70-100 neurons or more, spread out over a large spatial extent) and if each is firing spontaneously at 10-40 Hz, the tonic current is comprised of events at an aggregate rate of 700-4000 Hz. Even if the individual events were comparable to miniature events, this would seem to be a very large tonic current*.

*To clarify this, it would be straightforward to block spiking and measure the tonic current by washing in Gabazine. One could then compare the effects on spontaneous spiking of injecting a comparable negative current somatically in the absence of inhibition, or even inject a conductance if one wanted to get fancy*.

The reviewers raise a very important point that we agree deserved further attention in the manuscript. We have now added new data and a more detailed discussion supporting the main conceptual challenge to a divisive gain model: how is it that one can have a strongly coupled, tonically active network of GABAergic neurons that nonetheless have no effect on baseline spiking? We describe briefly the logic of the mechanism we believe can explain this empirical observation.

This question can be broken down in to two related questions: 1) Are the background rates of IPSCs consistent with the predicted convergence of collateral inputs? 2) How do SNr projection neurons appear to be relatively insensitive to this tonic background input?

Question 1) First, what is the predicted background rate of IPSCs? Our stimulation results, combined with an estimate of amplitude of a unitary synaptic connection, predicted 50B100 synapses from neighboring SNr projection neurons on to any given projection. If we assume that the lack of interneurons is correct this number is in good agreement with the ∼80 boutons per neuron estimated using bouton counting in reconstructions of the axon collaterals of projection neurons in the SNr of the rat (36) . We roughly agree with the reviewers that there could be a background rate of spiking in the range of 1B2 kHz (50B100 inputs * 20 Hz); although we note that this estimate is somewhat high given that in our attempts to measure the saturating number of synapses some connections were almost certainly the product of severed axons. Thus, in the slice the effective background rate of spiking will be somewhat reduced from predictions. The expected rate of IPSCs is then:

(1)RuIPSCs=Npre×Rpre×Prelease

Where R_uIPSCs_ is the predicted rate of unitary IPSCs (uIPSCs), N_pre_ is the number of presynaptic inputs (release sites), R_pre_ is the mean firing rate of presynaptic inputs, and P_release_ is the effective release probability at each release site. Although we, as in many other cases, do not have a direct measure of P_release_ it can be estimated. In general P_release_ is closer to 0 than to 1 for many central synapses. Likewise, our data are consistent with a low vesicular release probability combined with multiple release sites similar to that reported for Purkinje cell synapses by the Raman Lab (55). The most telling piece of data in this regard is the balance between vesicle recycling and vesicle release revealed by sustained stimulation. Despite the fact that SNr collateral synapses are depressing we found that even with sustained stimulation steady state depression is only ∼50B60% (very similar to that observed at Purkinje synapses; Figure 4—figure supplement 2.) This is consistent with a low vesicular release probability that allows vesicle recycling to keep pace with release and preventing complete depletion. If we assume release probabilities in the range of ∼0.1B0.3 (consistent with a binomial fit to the distribution of IPSCs observed in our voltage clamp experiments) we would obtain estimates of the expected background rate of uIPSCs in the range of 100B600 Hz, or less if some of the release sites were in severed axons or silent somata. We have now directly measured the background rate from spontaneous activity and observed a mean of ∼200 Hz (Figure 4, consistent with these predictions.

Question 2) Given a tonic background rate of IPSCs at ∼200 Hz this would predict that there is a tonic current less than the peak of a unitary event (τ_decay_ ≈ 5 ms). The question posed implies that tonic spiking is the problem. We would suggest that tonic spiking is, in this case, the solution. For a neuron to have tonic background firing it means that upon the return from a spike the net membrane current is constantly inward and thus drives the membrane towards threshold and the regenerative spike is triggered. This implies a positive slope conductance with a net inward current below threshold – in other words the biophysics of the neuron opposes hyperpolarizing currents in the perithreshold regime. This, combined with the reduced driving force of inhibition near threshold, suggests that tonic spiking neurons such as SNr projection neurons should be much less sensitive to inhibition relative to excitation. To test this hypothesis explicitly we performed dynamic clamp experiments in which we systematically varied the balance between a high background rate of inhibition and excitation. Indeed, we find that the sensitivity of SNr projection neurons to high background rates of inhibition is relatively weak compared with excitation (Figure 4). Indeed, for a tonic conductance equivalent to a uIPSC (∼150 pS) little or no change in spiking was observed. An alternative or complementary possibility is that homeostatic mechanisms could serve to counteract such changes. Thus, we also examined phase plots of the perithreshold membrane potential dynamics in the presence and absence of inhibition. We found that there was no change in any perithreshold or threshold properties (Figure 4—figure supplement 1). Moreover, we could directly observe the nonlinear kink in the perithreshold slope conductance such that near the inhibitory reversal potential there is a rather steep positive slope conductance that can oppose transient hyperpolarizing currents (i.e*.,* a background of uIPSCs).

Thus, taken together we believe our data show that the low release probability of the SNr synapses together with a net inward current and positive slope conductance in the perithreshold regime are sufficient to explain how a strongly coupled inhibitory neuron network of tonically spiking neurons can remain relatively unaffected by inhibition in the absence of stimulation yet strongly suppress changes in response to stimulation – i.e*.*, implement a divisive gain control mechanism. We propose that this suggests that evolution has discovered multiple implementations using inhibitory interneurons in other circuits, but in the basal ganglia appears to have discovered a mechanism in a circuit lacking interneurons.

*Potential concern about the electrotonic location of the feedback inhibition is mitigated by the fact that the events observed are rapid*.

The reviewers are quite right to point out that there are some potential issues with space clamp in these experiments. We have taken steps to mitigate space clamp issues (Cs^+^ based internal solution) and use nominal holding potentials that effectively isolate IPSCs (consistent with the estimated junction potential offset); however, space clamp is always imperfect. We have taken extra care to note this point in text.

*Although this proposed experiment might clarify the issue, it should be viewed as a suggestion and it may be feasible to clarify the issue with additional textual changes. Alternatively, it was also suggested that the manuscript would be improved by the inclusion of cell filling (at a minimum from slices but much better from in vivo staining) showing the axonal arborizations of the SNr projection neurons (this could also potentially address the concern above). If is not so, as the authors claim that the level of convergence of the SNr collaterals onto neighboring SNr (or DA) neurons cannot be reliably inferred from anatomical studies such as this, it would be more correct to say that this is beyond the scope of the present study, since many such studies have been performed in the laboratories of Somogyi, Bolam, Magill and others in the basal ganglia and elsewhere*.

While the abundance and organisation of feedback inhibition can be measured with anatomical studies, the functional properties of these synapses can still remain opaque. Our prediction of the density of synaptic contacts based on our functional measurements corresponds quite well to the anatomical data obtained from [36]. We do note that, to our knowledge, the correspondence between boutons and functional synapses has not yet been directly demonstrated at these collateral synapses and Mailly et al. measurements were made in rats not mice. Both points could be addressed by staining for multiple presynaptic protein markers and counting boutons in mice, however we believe this is beyond the scope of our current study.

*Finally, one reviewer felt that prior work supporting the idea that interneurons are absent in this structure should also be mentioned in the Discussion and not only raised in the Abstract*.

Thank you for this helpful comment. We have made the suggested change.